# Constitutive expression of the transcriptional co-activator IκBζ promotes melanoma growth and immunotherapy resistance

Antonia Kolb[1], Ana-Marija Kulis-Mandic[1], Matthias Klein[2,3], Anna Stastny[1], Maximilian Haist [1,4], Vanessa Votteler[5,6,7], Beate Weidenthaler-Barth[1], Tobias Sinnberg [7,8,9], Antje Sucker[10], Gabriele Allies[10], Lea Jessica Albrecht[10], Alpaslan Tasdogan [10], Andrea Tuettenberg[1,3], Matthias M. Gaida[3,11,12], Carsten Deppermann [3,13], Henner Stege[1], Dirk Schadendorf [10], Stephan Grabbe[1,3], Klaus Schulze-Osthoff[5,6,7] & Daniela Kramer [1,3] ✉

IκBζ, a rather unknown co-regulator of NF-κB, can either activate or repress a subset of NF-κB target genes. While its role as an inducibly expressed, transcriptional regulator of cytokines and chemokines in immune cells is established, IκBζ's function in solid cancer remains unclear. Here we show that IκBζ protein is constitutively expressed in a subfraction of melanoma cell lines, and around 30% of all melanoma cases, independently of its mRNA levels or known mutations. Deleting IκBζ in melanoma abrogates the activity and chromatin association of STAT3 and NF-κB, thereby reducing the expression of the pro-proliferative cytokines IL-1β and IL-6, thus impairing melanoma cell growth. Additionally, IκBζ suppresses *Cxcl9*, *Cxcl10*, and *Ccl5* expression via HDAC3 and EZH2, which impairs the recruitment of NK and CD8+ T cells into the tumor, causing resistance to α-PD-1 immunotherapy in mice. Thus, tumor-derived IκBζ may serve as a therapeutic target and prognostic marker for melanoma with high tumor cell proliferation, cytotoxic T- and NK-cell exclusion, and unfavorable immunotherapy responses.

IκBζ, encoded by *NFKBIZ*, belongs to the NF-κB co-factor family of atypical IκB proteins, which are known to induce or repress a specific subset of NF-κB target genes. Unlike classical IκBs, IκBζ is usually not constitutively expressed but rapidly induced upon activation of various immune cells, keratinocytes, or epithelial cells with toll-like receptor (TLR) ligands or pro-inflammatory cytokines, such as IL-1β, IL-17, or IL-36, in an NF-κB- and STAT3-dependent manner[1]. Depending on the stimulus and cell type, IκBζ protein levels are controlled by post-transcriptional regulation of its messenger RNA (mRNA) stability through three prime untranslated region (3′-UTR)-dependent processing[2,3], or by post-translational modifications, such as

phosphorylation or ubiquitination[4]. Upon protein expression, IκBζ localizes to the nucleus, where it mediates the induction of pro-inflammatory cytokines and chemokines, such as *IL1B, IL6, IL36, CXCL2*, or *CXCL8*[5]. Finally, IκBζ becomes degraded to balance the inflammatory response[2,3,5].

How IκBζ regulates gene expression is still not fully understood. Previously, it was shown that IκBζ can interact with transcription factors, such as p50, p52, or STAT3, on the chromatin, thereby possibly inhibiting or increasing their overall activation[6]. As IκBζ itself lacks any enzymatic activity, it has been suggested that chromatin-bound IκBζ acts as a bridging factor to recruit epigenetic modifiers to chromatin-

associated transcription factor complexes, thereby either facilitating or inhibiting the accessibility of specific promoter regions. Up to now, epigenetic modifiers such as TET2, SWI/SNF, or HDAC1 have been identified as interaction partners of IκBζ, although these interactions seem to be highly variable and dependent on the cell type and stimulus[7–9].

While the function of IκBζ in T cells or keratinocytes, for example, is well established by now[5,10], the role of IκBζ in cancer development and progression has rarely been investigated. So far, only in an aggressive subtype of diffuse large B-cell lymphoma (ABC-DLBCL), the expression and function of IκBζ have been studied in detail[11]. In this B-cell lymphoma subtype, IκBζ is constitutively expressed, leading to an increased expression of IL-6 and IL-10, which in turn enhance tumor growth and survival.

Besides DLBCL, overexpression of IκBζ has been detected in other hematological malignancies, including adult T-cell leukemia and mycosis fungoides, the most common subtype of primary, cutaneous T-cell lymphoma. However, the functional consequences of IκBζ expression in these diseases remain to be uncovered[12]. Moreover, the regulation and function of IκBζ expression in solid tumor entities, such as melanoma, are unknown.

In most cases, melanoma develops from ultraviolet (UV)-damaged melanocytes that acquire multiple mutations in proliferation-associated genes, such as *BRAF* or *NRAS*[13]. Historically, melanoma has been considered a highly lethal disease, as these tumors rapidly proliferate and metastasize to distal lymph nodes and organs. However, the invention of targeted therapies such as BRAF/MEK inhibitors and especially immunotherapy has significantly improved therapy options and overall survival rates for melanoma patients[14]. One commonly used mechanism of tumor immune evasion involves the expression of immune checkpoint proteins, such as PD-L1, within the tumor and myeloid-cell compartments. This expression limits effector T-cell responses through the activation of pathways involved in T-cell exhaustion, resulting in the upregulation of exhaustion markers, including PD-1 and LAG3[15]. Application of α-PD-1, α-PD-1L, or α-CTLA4 antibodies is blocking this immune-evasive mechanism, thereby reactivating cytotoxic T cells in the tumor microenvironment (TME) and inducing T-cell-dependent tumor cell death. Unfortunately, only 30-45% of melanoma patients respond to initial immunotherapy[16]. The mechanisms underlying immune checkpoint blockade (ICB) failure remain poorly understood and might be multifactorial. However, as ICB can only reactivate pre-existing cytotoxic T cells in the TME, the exclusion of T cells and natural killer (NK) cells from the tumor stroma has been identified as one major obstacle contributing to ICB resistance[17]. Understanding the mechanism underlying ICB failure might therefore identify therapeutic targets that could sensitize melanoma patients to immunotherapy approaches.

Here, we investigated the expression and function of IκBζ in melanoma using patient tumor samples, melanoma cell lines, and immunocompetent melanoma mouse models. We observed that a subfraction of melanoma exhibits constitutive IκBζ protein expression, which leads to enhanced STAT3 and NF-κB activation, thereby increasing the expression of pro-inflammatory cytokines and chemokines. At the same time, IκBζ repressed the expression of several chemokines, such as *Cxcl9* and *Cxcl10*, by promoting the recruitment of HDAC3 and EZH2 to the promoter regions of these genes. Consequently, IκBζ-dependent gene expression in melanoma not only sustains and enhances tumor cell proliferation but also modulates the cellular organization within the TME. This effect was especially evident for cytotoxic immune cells, as the constitutive IκBζ expression in melanoma suppressed their recruitment into the tumor stroma, thus inhibiting α-PD-1 antibody responses in vivo. In conclusion, we propose that constitutive IκBζ expression represents an attractive target for therapy approaches in melanoma, as inhibition of IκBζ can

suppress tumor growth and restore cytotoxic cell infiltration in the TME, thereby re-sensitizing tumors for immunotherapy.

## Results

### IκBζ is constitutively expressed in a subgroup of primary and metastatic melanoma

To study the role and functional implications of IκBζ expression in solid tumors, we chose malignant melanoma, which constitutes a highly immunogenic model tumor, thereby allowing insights into the role of IκBζ for shaping anti-tumor immune responses. In an initial discovery attempt, we investigated the mRNA and protein levels of IκBζ (encoded by *NFKBIZ*) in primary and metastatic human and murine melanoma cell lines, harboring diverse driver mutation profiles. While *NFKBIZ* mRNA was detectable in varying levels in all investigated melanoma cell lines, a subfraction of cell lines (including human melanoma cell lines LOX-IMVI, SK-MEL-30, and SK-MEL-5, or murine melanoma cell lines D4M-3A and YUMM1.7) displayed constitutive IκBζ protein expression (Fig. 1a). Of note, IκBζ protein levels did not correlate with its mRNA expression (Fig. 1a), with common mutations in *BRAF* or *NRAS* (Supplementary Fig. S1a), or with the activity of IκBζ-associated transcription factors such as NF-κB or STAT3 (Supplementary Fig. S1b). Instead, IκBζ protein levels, but not its mRNA levels, were effectively abrogated by treatment with 4EGI-1, an inhibitor of cap-dependent translation (Fig. 1b). Furthermore, inhibition of transcription using Actinomycin D decreased the mRNA levels of *NFKBIZ*, without affecting its protein levels (Supplementary Fig. S1c). Vice versa, inhibition of the proteasome using MG-132 restored IκBζ protein expression in melanoma cells that normally do not express IκBζ protein (Fig. 1c). Thus, constitutive expression of IκBζ protein in melanoma is governed by mechanisms related to post-transcriptional or post-translational regulation rather than transcriptional activity or typical melanoma mutations.

Next, we validated our findings in melanoma patient samples. Therefore, we confirmed IκBζ protein expression by immunohistochemistry (IHC) on formalin-fixed, paraffin-embedded (FFPE) samples from primary and metastatic melanoma, using a home-made antibody against human IκBζ, which we validated beforehand (Supplementary Figs. S1d, e). By analyzing IκBζ protein expression in 90 cases of malignant melanoma, we found that approximately 35% of all melanoma samples displayed IκBζ protein expression, which was uniformly distributed throughout the tumor (Fig. 1d). Moreover, we detected gradual changes in expression levels distinguishing moderate (15%) and high (20%) IκBζ expression, although its overall expression intensity did not correlate with tumor stage, or the anatomical site from which the tumor derived (Supplementary Figs. S1f, g). Importantly, and similar to melanoma cell lines, IκBζ protein expression did not correlate with its relative mRNA levels in patient samples (Supplementary Fig. S1h). Instead, melanoma patients with constitutive IκBζ protein expression in the tumor area were characterized by diminished progression-free survival (Fig. 1e).

### Constitutive IκBζ regulates the expression of melanoma-derived cytokines and chemokines

Next, we investigated the functional impact of constitutive IκBζ expression in melanoma. For this purpose, we used short hairpin RNA (shRNA) or CRISPR-Cas9 to knock down or knock out *NFKBIZ* (IκBζ) in the IκBζ-expressing LOX-IMVI and D4M-3A melanoma cells (Fig. 2a). Global transcriptome analysis of control and *NFKBIZ* knockdown or *Nfkbiz* knockout cells revealed a significant expression change of 527 genes in LOX-IMVI cells and 267 genes in D4M-3A cells (Fig. 2b), which were either up- or downregulated (with a minimum fold change of 2 and a *P* value ≤ 0.05). As expected, IκBζ-dependent pathways that were significantly enriched comprised the inflammatory response, as well as NF-κB and STAT3 signaling pathways (Fig. 2c). Furthermore, we found a significant overlap between IκBζ target genes in melanoma cells and a

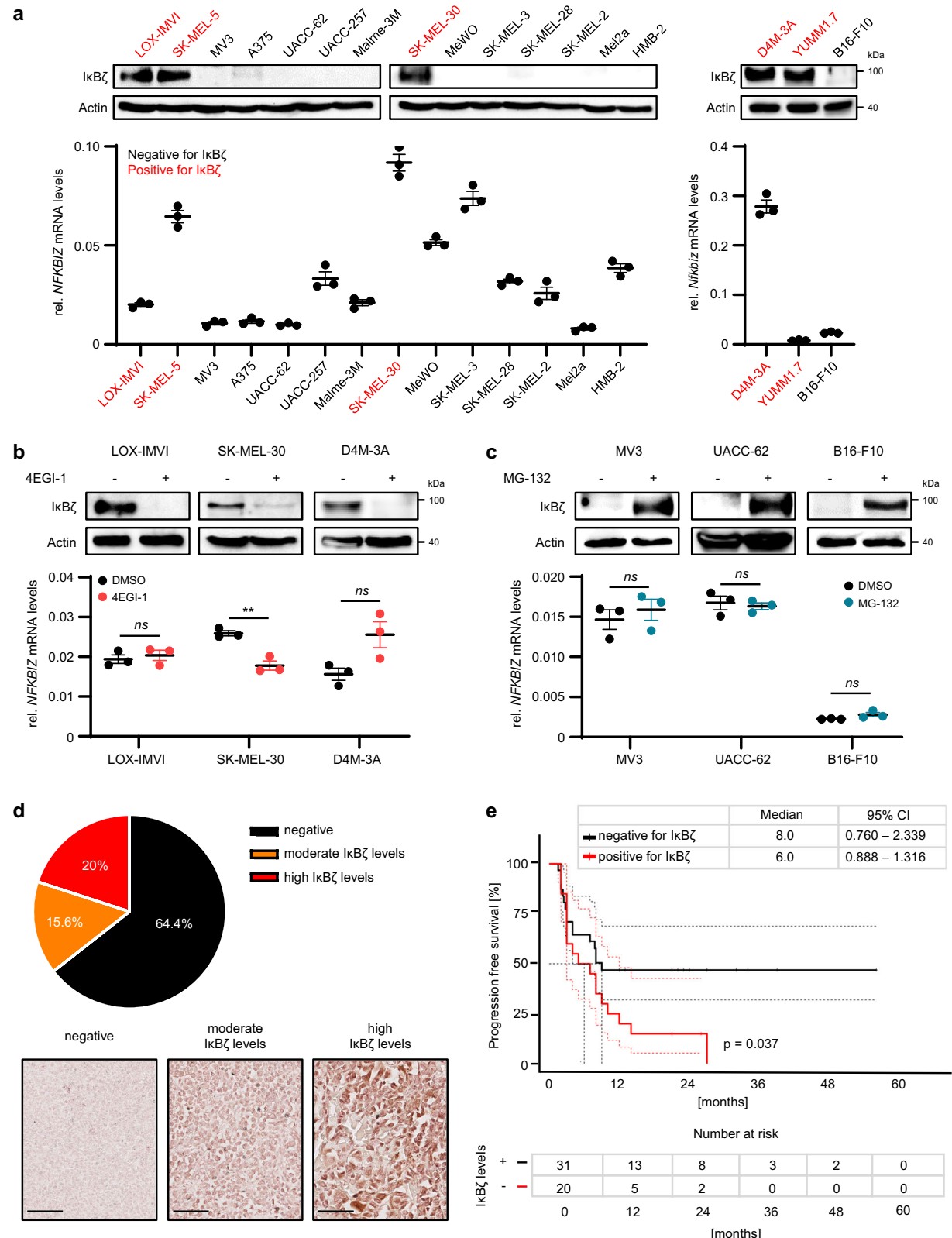

previously described gene set of inflammatory response-associated hallmarks, which became even more pronounced when applying a less stringent fold change and *P* value (Fig. 2d, e; and Supplementary Fig. 2b)[18]. Gene set enrichment analysis revealed that these IκBζ-dependent genes were associated with various biological functions, such as regulation of tumor growth, immune evasion, inflammation, metabolism, and survival. Furthermore, their IκBζ-dependency could

be validated across various melanoma cell lines, regardless of whether *NFKBIZ* was knocked down in cells with constitutive IκBζ expression, or transiently overexpressed in IκBζ-negative melanoma cells (Fig. 2f, g; and Supplementary Fig. S2a, and c–f). Finally, we detected direct binding of IκBζ to the respective target gene promoter sites by chromatin immunoprecipitation assays, indicating direct regulation of these target genes by IκBζ (Fig. 2h). Thus, we revealed that

**Fig. 1 | IκBζ protein is constitutively expressed in a subgroup of melanoma cell lines and melanoma patients. a**–**c** β-Actin serves as a loading control for immunoblot detection of IκBζ. Human *NFKBIZ* mRNA expression levels were normalized to *RPL37A*, and murine *Nfkbiz* mRNA expression to *Actb*, respectively. Shown is the mean of 3 independent experiments ± standard deviation (SD). Significance was calculated using a two-tailed Student's t-test (**\*\****p* < 0.01, *ns* = not significant). **a** IκBζ expression levels in various human and murine melanoma cell lines at steady state (Top: immunoblot detection, bottom: mRNA levels). **b** Detection of IκBζ (*NFKBIZ*) levels 24 h after treatment with 25 μM 4EGI-1 or a DMSO vehicle control. **c** IκBζ expression levels 4 h after treatment with 10 μM MG-132 or a DMSO vehicle control. **d** Relative amount of IκBζ-positive melanoma samples as detected by

immunohistochemical staining of FFPE samples. *Top:* Analysis of IκBζ protein expression in 90 metastatic melanoma patients using a commercial skin tissue array. *Bottom:* Representative pictures showing no expression of IκBζ, or moderate and high expression of IκBζ in melanoma. Scale bar: 100 μm. **e** Kaplan-Meier curve showing the progression-free survival (PFS) of 54 melanoma patients grouped according to the presence or absence of tumor-derived IκBζ protein expression. Shown is the median ± 95% CI. Significance was calculated using a two-tailed Student's t-test (\**p* < 0.05, \*\**p* < 0.01, and \*\*\**p* < 0.001). Survival analyses were conducted using the survminer R package (RStudio Version 1.3.1093). Source data and exact *P* values are provided in the Source Data file.

constitutively expressed IκBζ in melanoma constitutes a critical and conserved regulator of various cytokines and chemokines, including *IL1B, IL6, CCL5, CXCL9*, or *CXCL10*.

## Melanoma-derived IκBζ promotes self-sustained tumor cell proliferation in vitro

Next, we investigated the functional consequences of IκBζ and its target gene expression in melanoma. Previous studies suggested that IκBζ target genes, such as *IL1A, IL1B*, or *IL6*, promote tumor cell proliferation, although it is unclear whether these cytokines are predominantly produced by the tumor cells or the surrounding TME[19–21]. As the expression of these cytokines is strictly dependent on the presence of IκBζ in melanoma, we assessed whether IκBζ overexpression or knockout has a direct effect on tumor cell survival or proliferation. Interestingly, CRISPR-Cas9-mediated knockout of constitutive IκBζ in D4M-3A cells led to decreased cell viability (Fig. 3a). Conversely, overexpression of IκBζ did not change the proliferation and survival of the IκBζ-negative cell line MV3 under normal culture conditions (Supplementary Fig. S3a). However, upon serum withdrawal, IκBζ overexpression was sufficient to sustain tumor cell proliferation in the non-expressing melanoma cell lines B16-F10 and MV3, while control cells lacking IκBζ stopped proliferation and died (Fig. 3b, c). Importantly, the proliferation of fetal calf serum (FCS)-depleted control MV3 cells was restored by the addition of cell culture supernatant from IκBζ-overexpressing MV3 cells, supporting our hypothesis that tumor-derived cytokines drive FCS-independent tumor cell proliferation (Fig. 3d). Furthermore, supplementation of serum-depleted cell culture medium with IL-6 and IL-1β, two proliferation-promoting cytokines regulated by IκBζ, was sufficient to re-establish the proliferation of control MV3 cells under starvation conditions (Supplementary Fig. S3b).

## Melanoma-derived IκBζ promotes self-sustained tumor proliferation in vivo

We then analyzed whether the knockdown or overexpression of IκBζ in melanoma cells also impacted tumor cell proliferation and tumor growth in vivo using three well-defined murine melanoma cell lines, D4M-3A, YUMM1.7, and B16-F10[22–24]. After subcutaneous injection of control and IκBζ knockout D4M-3A or YUMM1.7 cells, or control and IκBζ-overexpressing B16-F10 cells into immunocompetent C57BL/6 mice, we monitored tumor growth over time. In agreement with our previous in vitro results, control D4M-3A and YUMM1.7 cells grew very fast, whereas IκBζ knockout D4M-3A and YUMM1.7 cells initially formed small palpable tumors 5–7 days after injection, but exhibited strong tumor growth inhibition thereafter (Supplementary Fig. S4a–c). Vice versa, overexpression of IκBζ in B16-F10 cells accelerated tumor growth compared to IκBζ-negative control B16-F10 cells, causing premature termination of the experiment on day 11, when IκBζ-overexpressing tumors started to ulcerate (Fig. 4a, b). Importantly, this tumor-promoting effect of IκBζ was TME-independent, as IκBζ-overexpressing B16-F10 tumors grew also faster in immunodeficient mice compared to the respective control cells (Fig. 4c). Molecular analysis of the tumors at the endpoint revealed diminished expression

of pro-proliferative cytokines in the IκBζ-deleted D4M-3A and YUMM1.7 tumors (Supplementary Fig. S4d), whereas IκBζ-overexpressing B16-F10 tumors displayed increased expression of pro-proliferative cytokines, such as *Il1a* or *Il6* (Fig. 4d). Thus, IκBζ expression in melanoma promotes tumor cell proliferation by increasing the expression of pro-proliferative cytokines.

In addition, we found that IκBζ simultaneously repressed several chemokines such as *Cxcl9, Cxcl10*, and *Ccl5* in D4M-3A, YUMM1.7, and B16-F10 melanoma in vivo (Fig. 4e; Supplementary Fig. S4d). Similar expression patterns of IκBζ target genes were confirmed in IκBζ-overexpressing B16-F10 tumors derived from immunodeficient mice, thereby validating that IκBζ-mediated gene expression occurs in a tumor-intrinsic manner also in vivo (Fig. 4f).

Due to this strong IκBζ-dependent repression of various chemokines in melanoma, we hypothesized that tumor-intrinsic IκBζ expression not only regulates tumor growth but also directly shapes the TME. To explore this, we investigated the composition of the TME of control and IκBζ-overexpressing B16-F10 tumors. Whereas most immune cell subtypes remained unchanged (Supplementary Fig. S4e), we detected a reduction of infiltrating CD3⁺ T cells in IκBζ-overexpressing B16-F10 tumors (Fig. 4g). Conversely, knockout of IκBζ increased the numbers of infiltrating CD3⁺ T cells into D4M-3A tumors (Supplementary Fig. S4f). Subsequent detailed analysis by flow cytometry revealed a significant decrease especially of tumor-infiltrating CD8⁺ T cells and NK cells in IκBζ-overexpressing B16-F10 tumors (Fig. 4h). In conclusion, IκBζ expression in melanoma inhibits tumor infiltration of cytotoxic T cells, probably by suppressing the expression of chemokines, such as *Ccl5, Cxcl9*, and *Cxcl10*.

## Constitutive IκBζ expression correlates with immunotherapy resistance in human melanoma patients

One key mechanism driving immunotherapy resistance involves the exclusion of cytotoxic T cells and NK cells from the tumor, due to the downregulation of important chemokines, such as *CCL5, CXCL9*, or *CXCL10*[25–27]. As constitutive IκBζ expression led to both significantly reduced levels of chemokines and decreased numbers of tumor-infiltrating cytotoxic T cells, we wondered whether IκBζ contributes to immunotherapy resistance, which is frequently observed in melanoma patients. Analysis of therapy-naïve melanoma patients revealed strong IκBζ staining, especially in patients who failed to respond to immunotherapy with nivolumab, either alone or in combination with ipilimumab (CR = complete response, PD = progression disease) (Fig. 5a; Supplementary Fig. S5a). Notably, this strong IκBζ expression was inversely correlated with the presence of tumor-infiltrating CD8⁺ T cells (Fig. 5b), predicting immunotherapy responses much more precisely than the tumor proportion score (TPS) of PD-1L-positive tumor cells[28] (Supplementary Fig. S5b). Importantly, we could rule out secondary effects of the immunotherapy on IκBζ expression, as most melanoma patients exhibited similar levels of IκBζ pre- and post-immunotherapy (Supplementary Fig. S5c). When IκBζ levels differed in patients before and after immunotherapy, we could only detect increasing IκBζ expression levels in patients exhibiting immunotherapy resistance, whereas some immunotherapy-sensitive

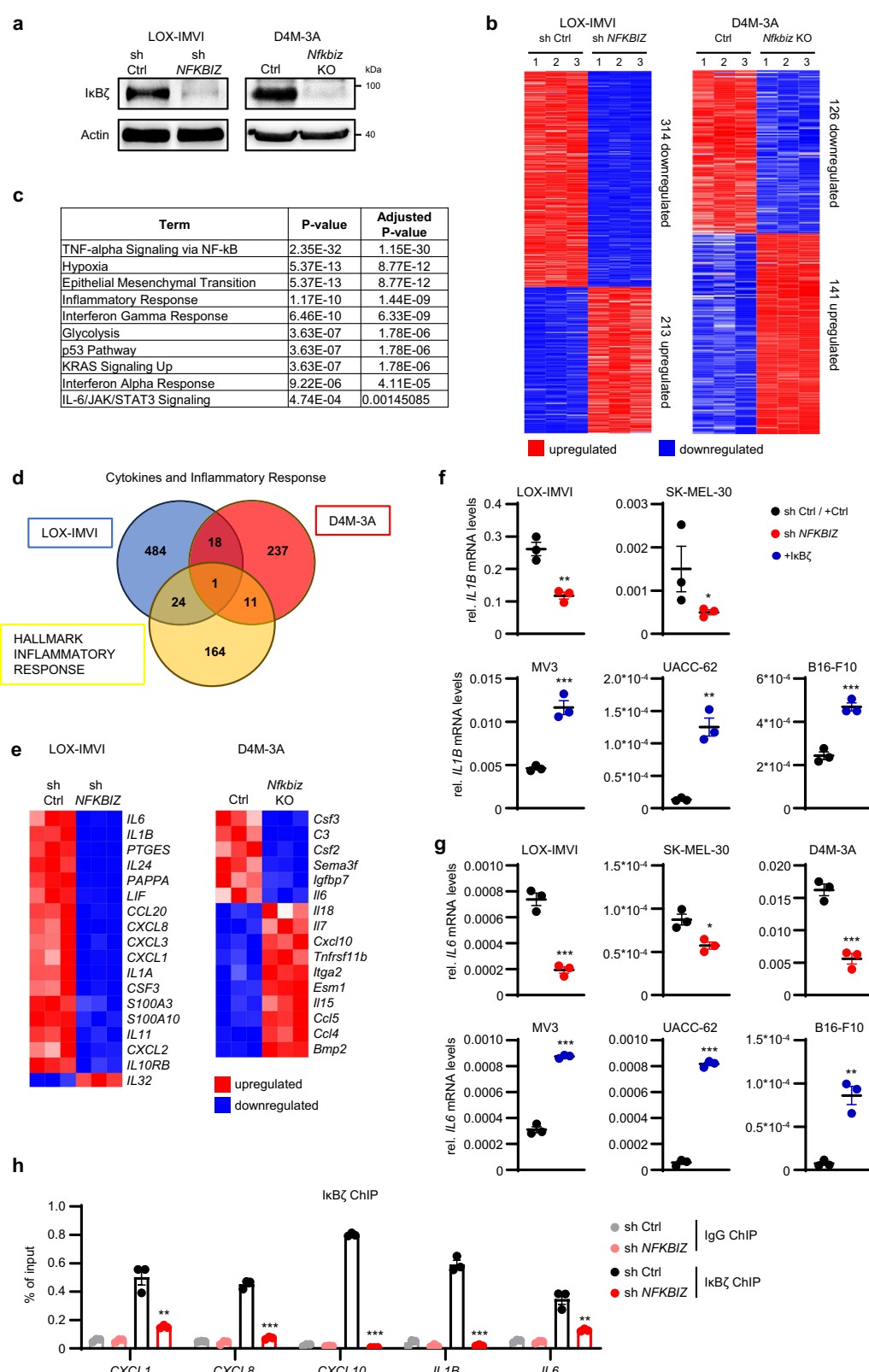

patients displayed remarkably reduced IκBζ levels. Thus, the presence of IκBζ protein in melanoma cells is strongly predictive of immunotherapy resistance. Based on these findings in human patients, we suggest that constitutive IκBζ expression actively induces immunotherapy resistance by suppressing chemokine expression, thereby inhibiting the recruitment of cytotoxic T cells into the tumor stroma.

## Constitutive overexpression of IκBζ in B16-F10 cells promotes immunotherapy resistance in vivo

To test this hypothesis, we assessed therapy responses in immunocompetent mice harboring control and IκBζ-overexpressing B16-F10 tumors after intraperitoneal injections of either IgG or α-PD-1 antibodies (Fig. 6a). While control B16-F10 tumors showed consistent cell growth inhibition upon α-PD-1 treatment, IκBζ-overexpressing B16-

**Fig. 2 | IκBζ regulates the expression of cytokines and chemokines in melanoma cells. a** IκBζ knockdown and knockout in LOX-IMVI and D4M-3A cells, respectively. IκBζ was knocked down or deleted by lentiviral transduction using either control or *NFKBIZ* shRNA, or an empty control and a *Nfkbiz* CRISPR-Cas9 construct. Efficient knockdown was validated by immunoblot detection of IκBζ, normalized to β-Actin. **b** Heatmaps showing all genes that were either up- (red) or downregulated (blue) upon depletion of IκBζ in LOX-IMVI or D4M-3A cells. Each condition included 3 independent experiments. IκBζ-regulated genes were defined based on a minimum fold change of 2, and *p* ≤ 0.05 (D4M-3A) or *p* ≤ 0.06 (LOX-IMVI). **c** Selected results from a gene set enrichment analysis of IκBζ target genes in LOX-IMVI cells using Enrichr (MSigDB Hallmark 2020)[56]. **d** Comparison of IκBζ target genes in LOX-IMVI or D4M-3A cells encoding for cytokines and chemokines,

with a previously published gene set of inflammatory target genes (GSEA term hallmark inflammatory response)[18]. **e** Detailed heatmap of the RNA sequencing results from **b** showing all IκBζ-regulated cytokines and chemokines. **f** + **g** Relative expression of *IL1B* and *IL6* in various *NFKBIZ* knockdown or IκBζ-overexpressing melanoma cells. Relative mRNA levels were normalized to the reference gene *RPL37A* (human) or *Actb* (murine). **h** Chromatin immunoprecipitation (ChIP) assays of IgG (as control) and IκBζ in shRNA control and *NFKBIZ* knockdown LOX-IMVI cells. Data represent the mean of 3 independent experiments ± standard deviation (SD). Significance was calculated using a two-tailed Student's t-test (*\*p* < 0.05, *\*\*p* < 0.01, and *\*\*\*p* < 0.001). Source data and exact *P* values are provided in the Source Data file.

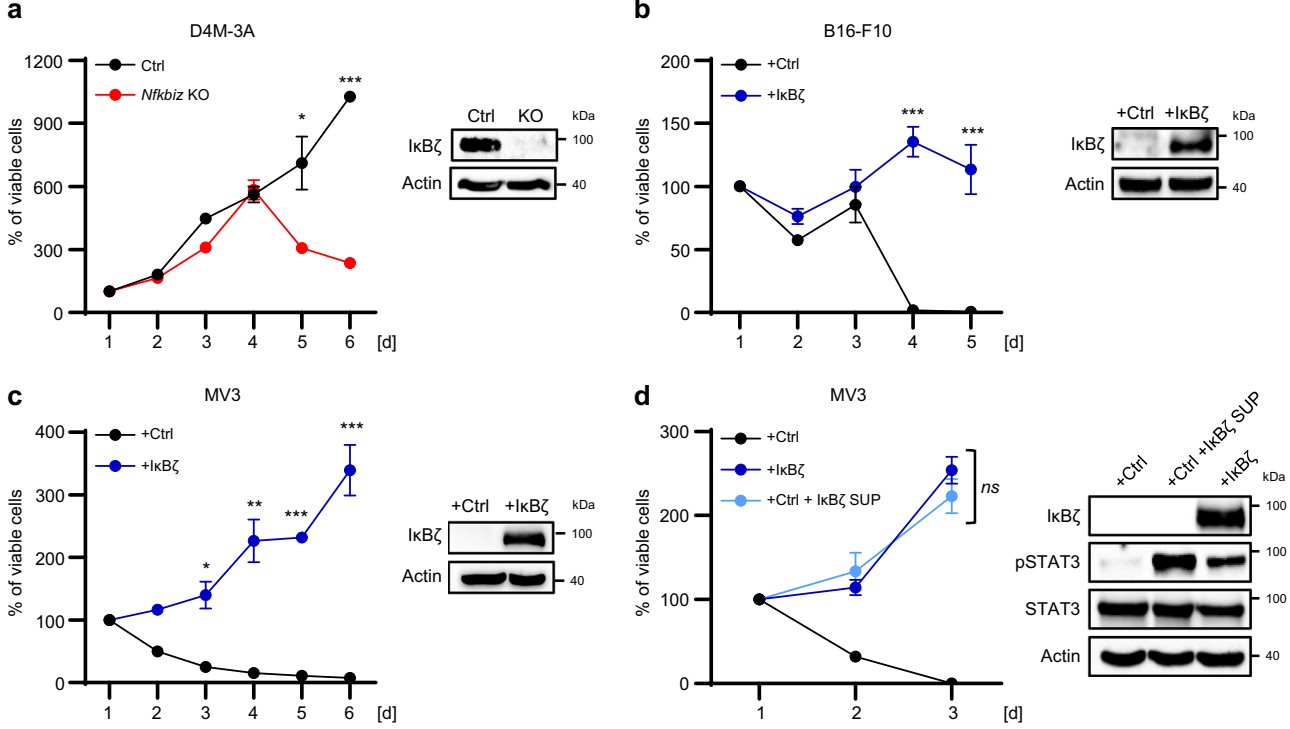

**Fig. 3 | Melanoma-derived IκBζ promotes self-sustained cell proliferation and tumor growth in vitro. a–d** In vitro cell proliferation was assessed using the CellTiter-Glo assay (Promega). For calculation, values obtained 24 h after initial seeding were set to 100%, and follow-up measurements were calculated as the percentage relative to the values from the first time point. IκBζ overexpression or knockdown, or phosphorylation of STAT3 (Y705) were analyzed by immunoblotting and normalized to β-Actin for each experiment. **a** Control or *Nfkbiz* knockout D4M-3A cells cultured in FCS-containing medium. **b** Control or IκBζ-

overexpressing B16-F10 cells, cultured under starvation conditions (without FCS). **c** Control or IκBζ-overexpressing MV3 cells, cultured under starvation conditions (without FCS). **d** Same as in (**c**) with control cells cultured in the presence of supernatant from IκBζ-overexpressing MV3 cells that were previously cultured without FCS. Cell proliferation assay show the mean of 3 independent experiments ± standard deviation (SD). Significance was calculated using a two-tailed Student's t-test (*\*p* < 0.05, *\*\*p* < 0.01, and *\*\*\*p* < 0.001, *ns* = not significant). Source data and exact *P* values are provided in the Source Data file.

F10 tumors retained tumor growth (Fig. 6b, c). This was not due to changes in the level of IκBζ overexpression, as application of α-PD-1 antibodies did not alter IκBζ expression (Fig. 6d). Instead, α-PD-1 treatment of control B16-F10 tumors increased the infiltration of immune cells, especially CD8+ T cells and NK cells, which was undetectable in IκBζ-overexpressing tumors (Fig. 6e, f; and Supplementary Fig. S6a, b). Moreover, we detected increased expression of *Ccl5*, *Cxcl9*, and *Cxcl10* in α-PD-1-treated control B16-F10 tumors, whereas IκBζ overexpression in B16-F10 tumors suppressed the expression of these chemokines (Fig. 6g). In agreement with these results, immunofluorescence analysis confirmed a strong infiltration of CD45+ immune cells together with elevated CXCL9 expression in α-PD-1-treated control B16-F10 tumors, which was strongly inhibited by IκBζ

overexpression, and derived from both tumor and infiltrating CD45+ immune cells (Fig. 6h).

### Knockout of endogenous IκBζ expression sensitizes YUMM1.7 cells to immunotherapy

We next explored whether the loss of endogenous IκBζ could re-sensitize to immunotherapy. For this purpose, we subcutaneously injected control and *Nfkbiz* knockout (KO) YUMM1.7 cells into C57BL/6 mice and treated the mice with IgG or α-PD-1 antibodies when the tumors became palpable (Fig. 7a, b). As previously reported, parental YUMM1.7 cells are resistant to immunotherapy[24]. However, knockout of IκBζ alone already severely reduced the tumor growth of YUMM1.7 cells, which was slightly but significantly

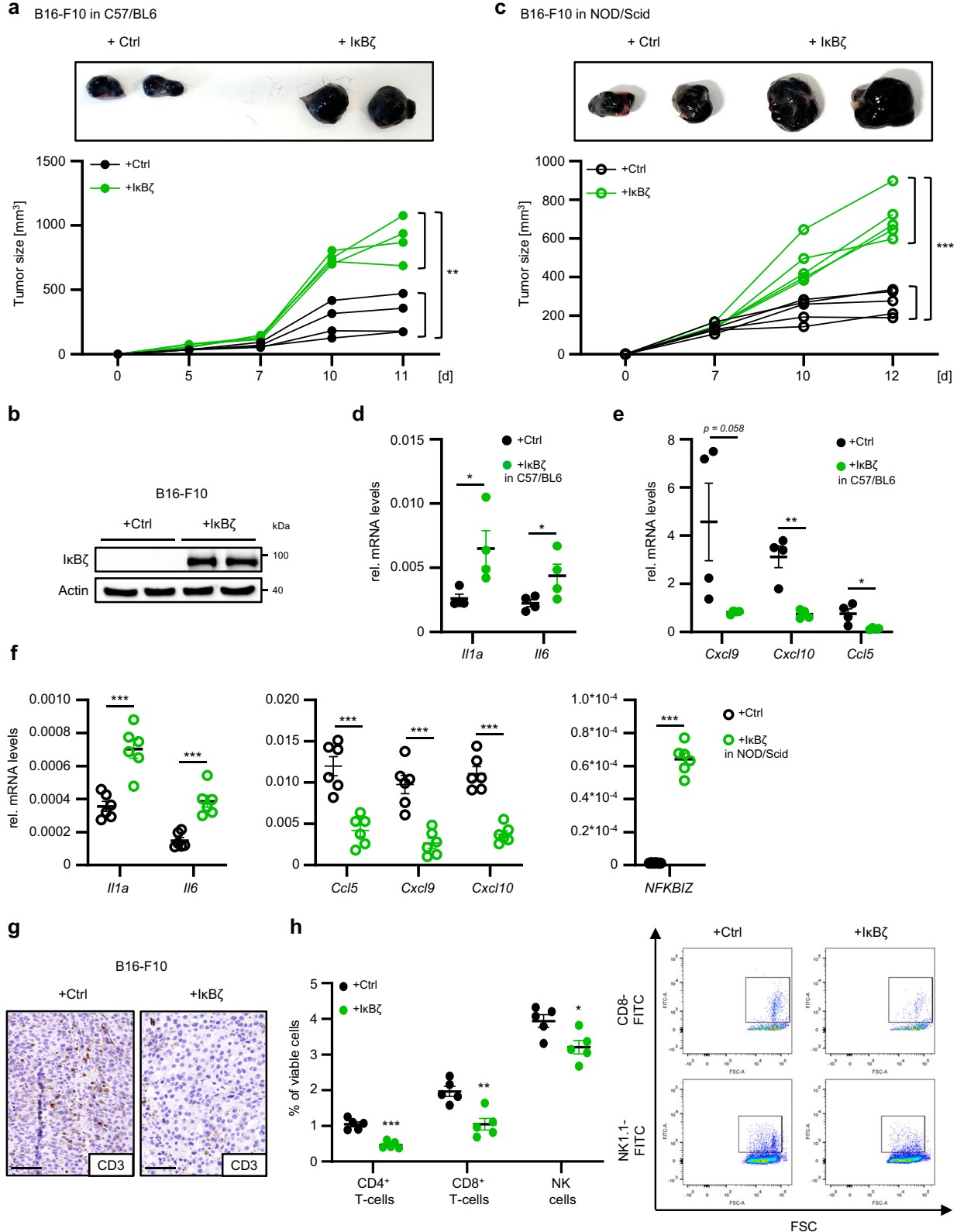

enhanced by the treatment with α-PD-1 antibodies (Fig. 7c). Subsequent flow cytometry analysis showed a mild increase in the recruitment of dendritic cells, monocytes, CD8⁺ T cells and NK cells upon IκBζ depletion. The immune cell infiltration was further intensified by α-PD-1 treatment of mice harboring IκBζ KO tumors, and absent in the IgG- and α-PD-1-treated control group (Fig. 7d). Importantly, re-establishment of an anti-tumorigenic TME in α-PD-1-

treated *Nfkbiz* KO YUMM1.7 tumors was accompanied by a significant increase in the expression levels of *Ccl5*, *Cxcl9*, and *Cxcl10* (Fig. 7e). Together this data confirms that constitutive IκBζ expression in melanoma contributes to immunotherapy resistance, supposably by suppressing *Ccl5*, *Cxcl9*, and *Cxcl10* expression, leading to a continuous exclusion of cytotoxic T cells and NK cells from the TME.

**Fig. 4 | Melanoma-derived IκBζ promotes self-sustained cell proliferation and tumor growth in vivo. a** Tumor growth of control and IκBζ-overexpressing B16-F10 cells, which were subcutaneously injected in the left and right flanks of C57BL/6 mice. $n = 4$. **b** Immunoblot confirming IκBζ overexpression in B16-F10 cells at the endpoint of the experiment. β-Actin staining served as a loading control. **c** Tumor growth of control and IκBζ-overexpressing B16-F10 cells, which were subcutaneously injected in the left and right flanks of NOD/Scid mice. $n = 5$. **d** + **e** Relative gene expression levels of IκBζ target genes in control and IκBζ-overexpressing B16-F10 tumors from C57BL/6 mice at the endpoint, normalized to *Hprt1*. $n = 4$. **d** Pro-proliferative cytokine expression. **e** Chemokine expression. Relative mRNA levels were normalized to the reference gene *Actb*. **f** Relative gene expression levels of cytokines and chemokines in control and IκBζ-overexpressing B16-F10 tumors from NOD/Scid mice at the endpoint, normalized to *Hprt1*. $n = 6$. **g** Immunohistochemical staining of CD3$^+$ T cells in control or IκBζ-overexpressing B16-F10 tumors from C57BL/6 mice at the endpoint. Scale: 100 μM. **h** Flow cytometry analysis of infiltrating T cells into control or IκBζ-overexpressing B16-F10 tumors from C57BL/6 mice. $n = 5$. The following markers were applied on living (DAPI negative) cells: cytotoxic CD8$^+$ T cells = CD3$^+$ CD8$^+$; CD4$^+$ T cells = CD3$^+$ CD4$^+$ CD25$^+$; NK cells = CD3$^-$ NK1.1$^+$. Shown is the mean ± standard error mean (SEM). Significance was calculated using a two-tailed Student's t-test (*$p < 0.05$, **$p < 0.01$, ***$p < 0.001$, ns = not significant). Source data and exact $P$ values are provided in the Source Data file.

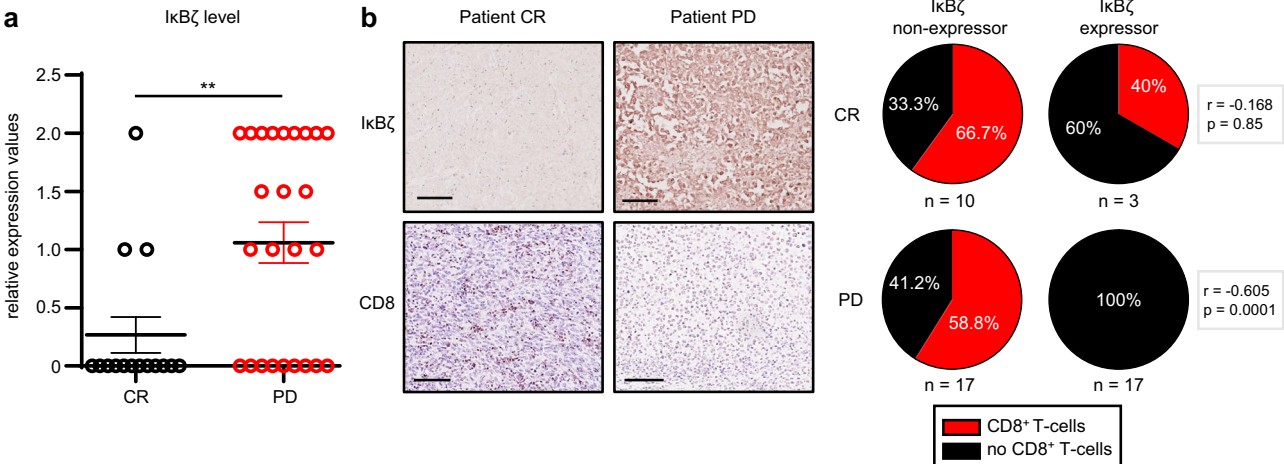

**Fig. 5 | IκBζ expression in human melanoma is associated with immunotherapy resistance and inversely correlates with CD8$^+$ T-cell infiltration. a** Relative protein levels of tumor cell-derived IκBζ in melanoma samples from patients with different responses to immunotherapy with nivolumab and/or ipilimumab. CR = complete response, PD = progressive disease. No detectable IκBζ expression was graded as 0, moderate expression as 1, and strong expression as 2. $n = 15$ (for CR) or $n = 25$ (for PD). Significance was calculated using a two-tailed Student's t-test (**$p < 0.01$). **b** *Left:* Representative immunohistochemical images demonstrating an inverse correlation of infiltrating CD8$^+$ T cells and IκBζ expression in two melanoma samples from an immunotherapy-sensitive (CR) and resistant (PD) patient. *Right:* Correlation analysis of tumor-infiltrating CD8$^+$ T cells and IκBζ expression in patients with different immunotherapy responses (CR vs. PD). Spearman's rank correlation coefficient r and corresponding two-tailed $P$ values were calculated (CR: $r = -0.168$; $p = 0.85$ and PD: $r = -0.605$; $p = 0.0001$). Black = absence of CD8$^+$ T cells, red = presence of CD8$^+$ T cells. Source data and exact $P$ values are provided in the Source Data file.

## Constitutive IκBζ expression in melanoma enhances the activity and chromatin association of p65, p50, and STAT3 to induce pro-proliferative cytokines

Finally, we aimed to understand how IκBζ regulates the expression of cytokines and chemokines in melanoma cells. As IκBζ cannot bind DNA directly, it is thought to interact with specific transcription factors, such as NF-κB or STAT, at the chromatin, thereby changing promoter accessibility by the recruitment of epigenetic modifiers[1]. NF-κB, STAT1, and STAT3 are key regulators of the previously identified IκBζ target genes, such as *IL1B* and *IL6*. Hence, we closely investigated their functional activity in IκBζ-expressing and non-expressing melanoma cells. We first assessed their overall activity by analyzing the relative phosphorylation levels of STAT1 (Y701), STAT3 (Y705), and IκBα (S32). In vitro, *NFKBIZ* knockout in YUMM1.7 cells or knockdown in LOX-IMVI cells reduced phosphorylation levels of STAT3 and STAT1, whereas overexpression of IκBζ in B16-F10 cells increased STAT3 phosphorylation (Fig. 8a). Notably, IκBα levels, as a marker for NF-κB activity, remained largely unaffected (Supplementary Fig. S7a). In agreement, in vivo analysis detected reduced phosphorylation of STAT3 (Y705) in *Nfkbiz* knockout D4M-3A tumors, whereas IκBζ-overexpressing B16-F10 tumors displayed increased levels of phosphorylated STAT3 (Supplementary Fig. S7b). Finally, pSTAT3 levels also strongly correlated with constitutive IκBζ expression in human melanoma samples, especially in immunotherapy-resistant patients, validating the IκBζ-dependent regulation of STAT3 activation in melanoma (Fig. 8b).

Next, we investigated the activity of the transcription factors by analyzing their chromatin localization in the presence or absence of IκBζ. STAT1, STAT3, and the NF-κB subunits p65 and p50 partially localized to the chromatin in wild-type LOX-IMVI cells with constitutive IκBζ expression, whereas knockdown of *NFKBIZ* almost completely abolished the association of these transcription factors with the chromatin fraction (Fig. 8c). Chromatin immunoprecipitation (ChIP) analyses further revealed specific binding of STAT3, p65, and p50 to the promoter sites of previously identified IκBζ target genes (such as *IL6* or *IL1B*), which was completely lost upon *NFKBIZ* knockdown (Fig. 8d; and Supplementary Fig. S8c). Together these data indicate that IκBζ-dependent gene expression is potentially mediated by its regulation of STAT1/3, p65, and p50 transcription factor function.

To further test this hypothesis, we overexpressed IκBζ in MV3 cells and analyzed IκBζ target gene expression in MV3 cells lacking either STAT3, p65, or STAT1. Consistent with the previous findings, IκBζ overexpression enhanced the expression of several pro-inflammatory genes, such as *IL6*, *CXCL1*, and *CXCL8*, whereas the knockdown of STAT3 abrogated these effects (Fig. 8e). Similarly, knockdown of p65 suppressed IκBζ-dependent upregulation of *IL1B*, *IL6*, and *CXCL8*, whereas STAT1 knockdown was sufficient to inhibit IκBζ-dependent upregulation of *IL6* and *CXCL8* in MV3 cells (Supplementary Fig. S7d, e). Consequently, inhibiting STAT1/3 or NF-κB using the STAT inhibitor Stattic or the IKK inhibitor IMD-0534, also

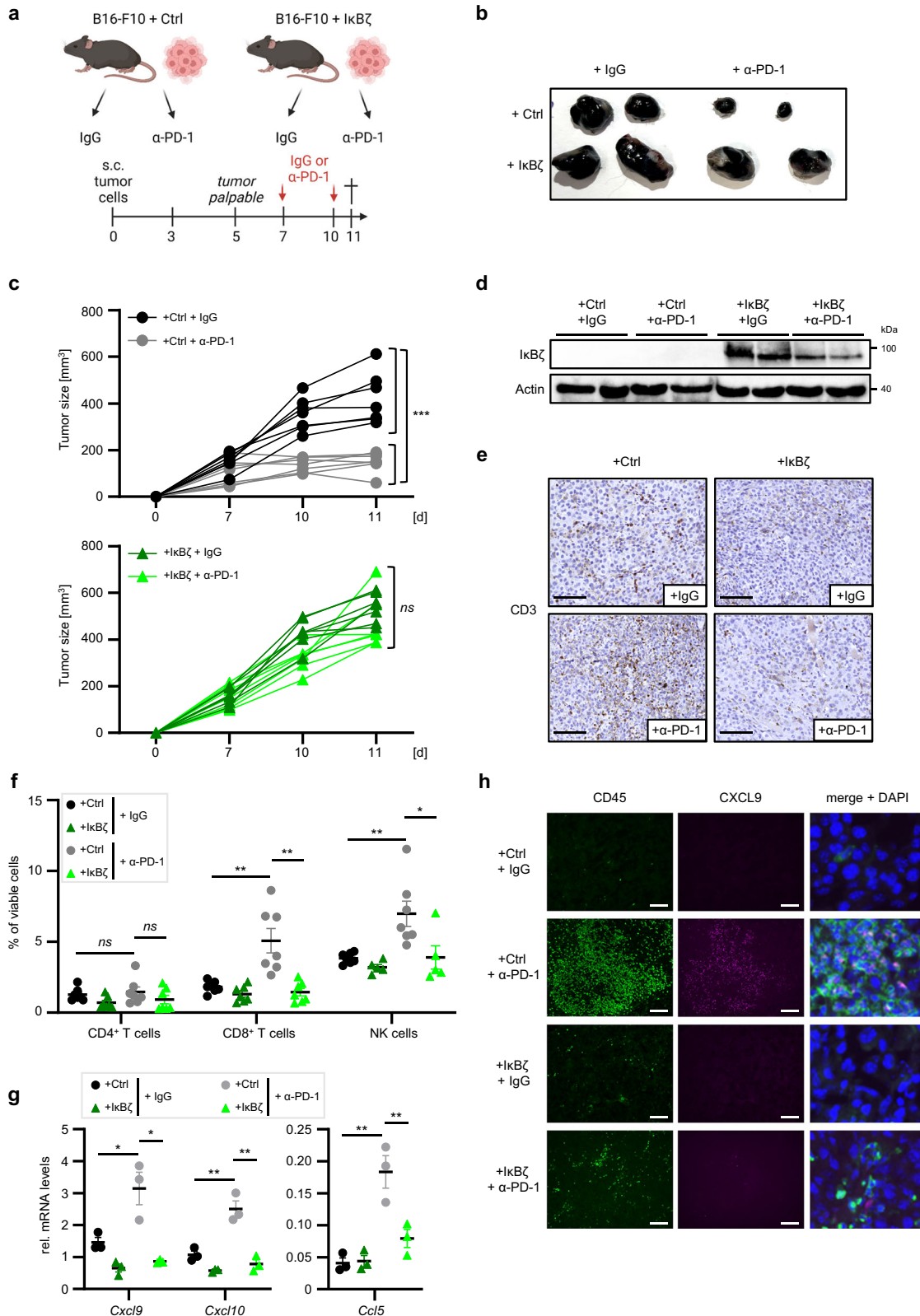

suppressed IκBζ-dependent, self-sustained tumor cell proliferation in MV3 cells (Supplementary Fig. S7f). In summary, constitutively expressed IκBζ in melanoma stabilizes specific promoter binding and chromatin association of STAT3, p65, and potentially STAT1, depending on the specific target gene. This enhanced interaction leads to increased STAT3 and NF-κB activity, thereby promoting tumor cell proliferation through IL-1 and IL-6.

## IκBζ recruits EZH2 and HDAC3 to mediate the repression of *CCL5*, *CXCL9*, and *CXCL10* in melanoma cells

Although increased STAT3 and NF-κB activity in IκBζ-expressing melanoma could explain the elevated expression of IL-6 and IL-1 cytokines, STAT3 and NF-κB inhibition was unable to recover IκBζ-mediated suppression of *Cxcl9*, *Cxcl10*, and *Ccl5* in melanoma cells (Supplementary Fig. S8a, b). Therefore, we concluded that this IκBζ-dependent

**Fig. 6 | Overexpression of IκBζ in B16-F10 promotes immunotherapy resistance. a** Scheme of the mouse experiment, assessing the response of control and IκBζ-overexpressing B16-F10 tumors to α-PD-1 immunotherapy. Created in BioRender. Kramer, D. (https://BioRender.com/e11y470). **b** Representative pictures of the tumors of all 4 experimental groups at day 11 (endpoint). **c** Tumor growth over time in IgG or α-PD-1 antibody-treated animals with either control B16-F10 tumors (top), or IκBζ-overexpressing B16-F10 tumors (bottom). *n* = 7 tumors per group. **d** Immunoblot analysis of IκBζ overexpression in the tumor material at the endpoint, normalized to β-Actin. n = 2 tumors per group. **e** Immunohistochemistry to detect CD3⁺ T cells in IgG- or α-PD-1-treated control

or IκBζ-overexpressing B16-F10 tumors at the endpoint. Scale: 100 µm. **f** Flow cytometry analysis of tumor-infiltrating lymphocytes, gated on living cells. The following markers were applied: cytotoxic CD8⁺ T cells = CD3⁺ CD8⁺; CD4⁺ T cells = CD3⁺ CD4⁺ CD25⁻; NK cells = CD3⁻ NK1.1⁺. *n* = 7. **g** Gene expression of *Cxcl9*, *Cxcl10*, and *Ccl5* in the tumors at endpoint, normalized to *Hprt1*. *n* = 3. **h** Immunofluorescence staining of CD45 and CXCL9 in B16-F10 tumors. The presented data derive from biological replicates ± standard error mean (SEM). Significance was calculated using a two-tailed Student's t-test (*$p < 0.05$, **$p < 0.01$, and ***$p < 0.001$, *ns* = not significant). Source data and exact *P* values are provided in the Source Data file.

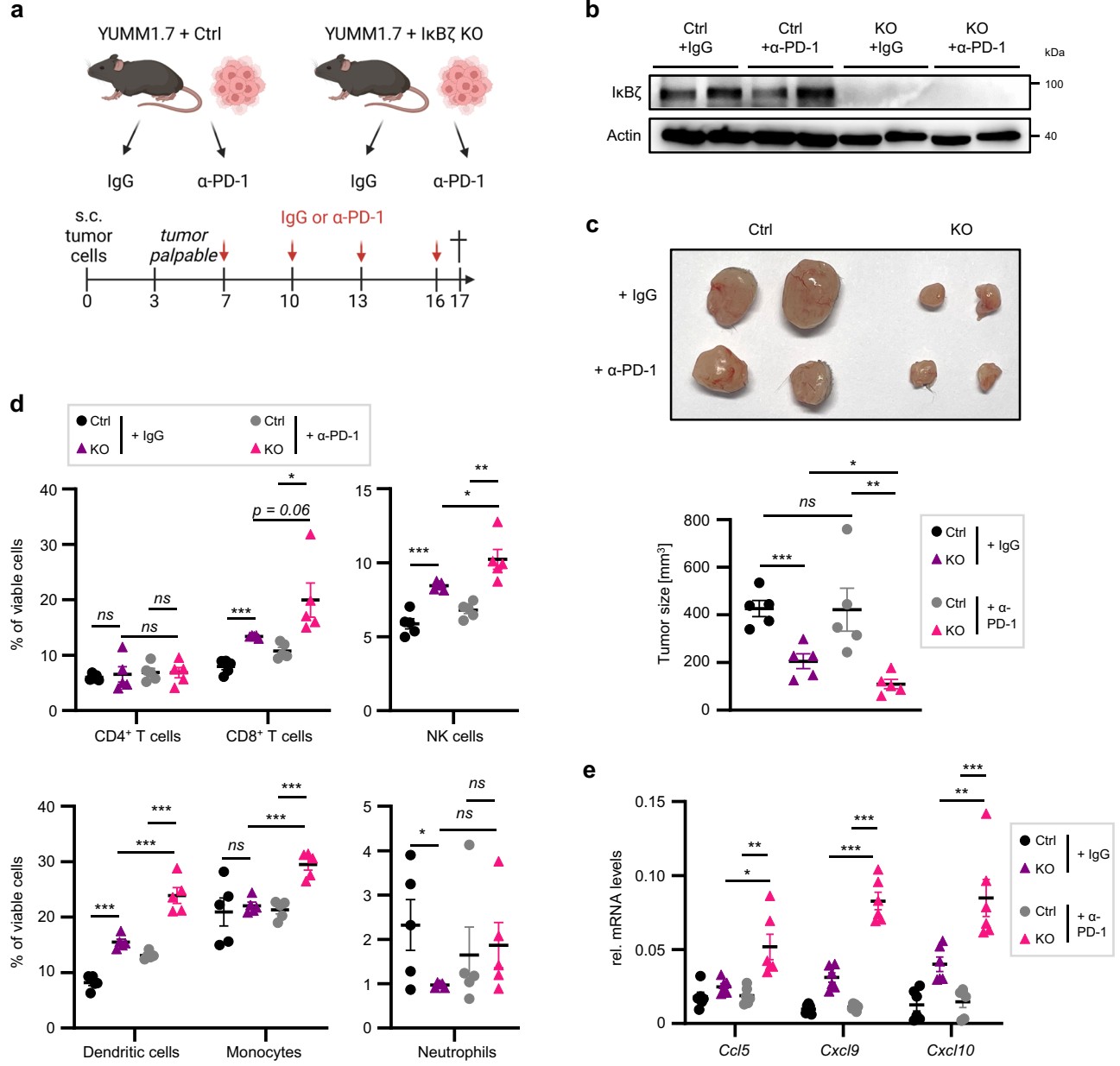

**Fig. 7 | Knockout of IκBζ in YUMM1.7 cells re-sensitizes to immunotherapy. a** Scheme of the mouse experiment assessing the response of control and IκBζ-depleted YUMM1.7 tumors to α-PD-1 immunotherapy. Created in BioRender. Kramer, D. (https://BioRender.com/) k58w881. **b** Immunoblot analysis of IκBζ knockout in the tumor material at the endpoint, normalized to β-Actin. *n* = 2 tumors per group. **c** *Top:* Representative picture of the tumors of all 4 experimental groups at day 17 (endpoint). *Bottom:* Tumor size at the endpoint in IgG- or α-PD-1 antibody-treated animals with either control or IκBζ-depleted YUMM1.7 tumors. **d** Flow cytometry analysis of tumor-infiltrating lymphocytes, gated on living cells. The

following markers were applied: cytotoxic CD8⁺ T cells = CD3⁺ CD8⁺; CD4⁺ T cells = CD3⁺ CD4⁺ CD25⁻; NK cells = CD3⁻ NK1.1⁺; Monocytes = Ly6C⁺; Neutrophils = Ly6G⁺ and dendritic cells = CD11c⁺. **e** Gene expression of *Cxcl9*, *Cxcl10*, and *Ccl5* in the tumors at the endpoint, normalized to *Actb*. Represented data derive from 5 biological replicates ± standard error mean (SEM). Significance was calculated using a two-tailed Student's t-test (*$p < 0.05$, **$p < 0.01$, and ***$p < 0.001$, *ns* = not significant). Source data and exact *P* values are provided in the Source Data file.

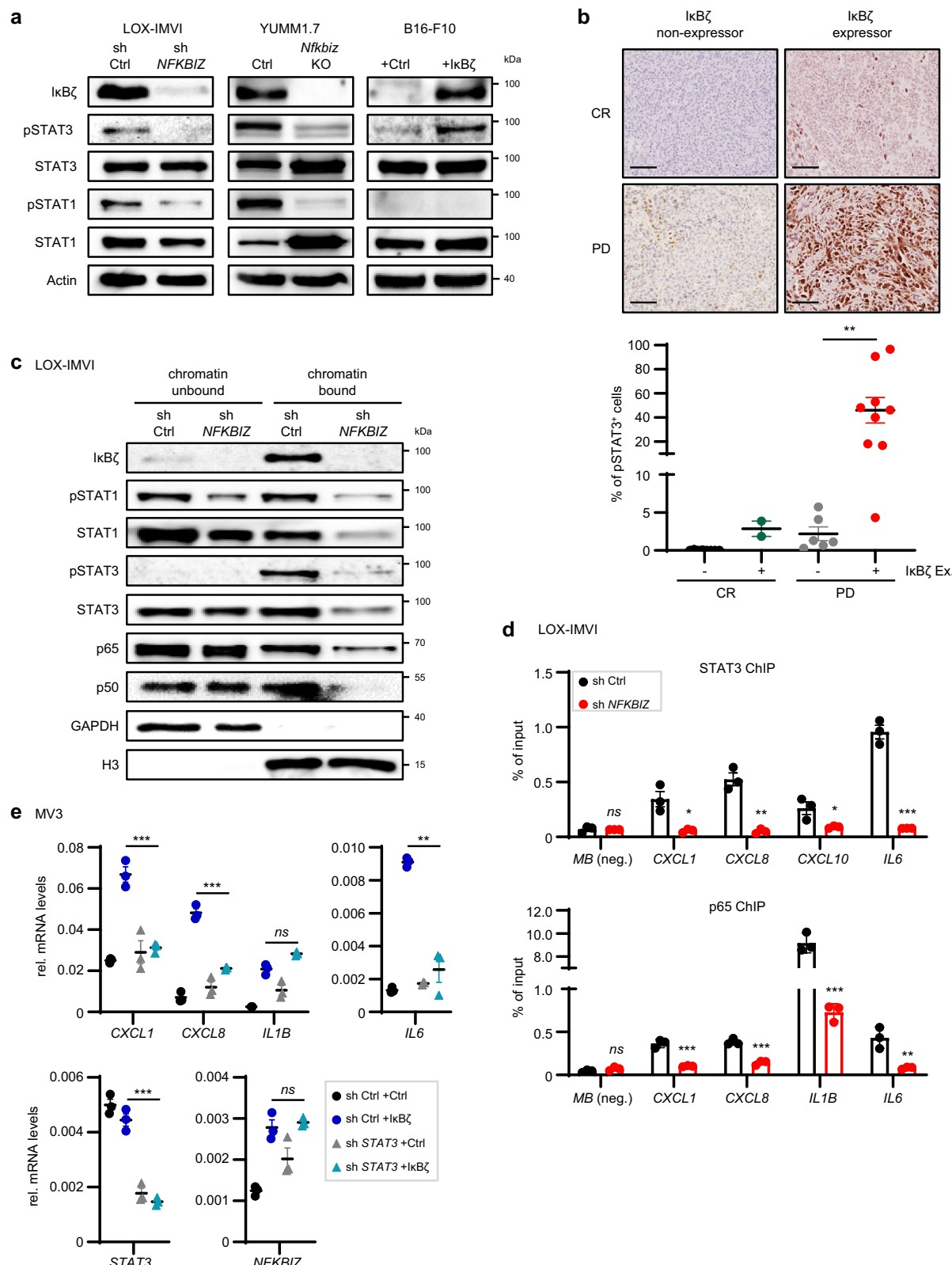

repression of these chemokines is mediated through an alternative mechanism.

Previous studies established that the epigenetic modifier EZH2, which mediates H3K27 trimethylation and heterochromatin formation, can specifically repress the expression of *Cxcl9* and *Cxcl10* in several tumor entities, leading to the exclusion of cytotoxic T cells and NK cells from the TME[29,30]. Furthermore, HDAC3, as a part of the NCoR

repressor complex, can suppress the expression of *Cxcl10* and *Ccl5* in several tumor entities, leading to T-cell and NK-cell exclusion from the TME and immunotherapy resistance[31,32]. Notably, inhibition of both EZH2 or HDAC3, can restore adaptive immunity in the TME and re-sensitize melanoma to immunotherapy[33,34].

Importantly, we previously demonstrated that keratinocyte-derived IκBζ can interact with HDACs, thereby mediating gene

**Fig. 8 | IκBζ regulates the activity and chromatin association of STAT1, STAT3, and p65 in melanoma. a** Immunoblot analysis of total and phosphorylated STAT3 (Y705) and STAT1 (Y701), in control or IκBζ-depleted LOX-IMVI or YUMM1.7 cells, and in control or IκBζ-overexpressing B16-F10 cells. **b** IHC staining of phosphorylated STAT3 (Y705) in human melanoma samples. *Top:* Exemplified pictures of pSTAT3 staining in IκBζ-expressing and non-expressing tumors from patients showing either a complete response (CR) or progressive disease (PD) upon immunotherapy. *Bottom:* Correlation of the presence of IκBζ protein and pSTAT3 levels in tumor cells of tissue sections. **c** Chromatin fractionation of LOX-IMVI control or *NFKBIZ* knockdown cells. GAPDH and histone H3 served as internal controls for the chromatin-unbound and chromatin-bound fractions, respectively.

**d** Chromatin immunoprecipitation of STAT3 and p65 in control or *NFKBIZ* knockdown LOX-IMVI cells. **e** Gene expression of IκBζ target genes in control or IκBζ-overexpressing MV3 cells, in the presence or absence of STAT3. *STAT3* was lentivirally knocked down using shRNA, and control cells were generated using a non-coding shRNA (sh Ctrl). Gene expression was subsequently analyzed in the presence or absence of transient IκBζ overexpression and normalized to the reference gene *RPL37A*. Data derived from 3 independent experiments (mean ± standard deviation (SD)). Significance was calculated using a two-tailed Student's t-test (*$p < 0.05$, **$p < 0.01$, and ***$p < 0.001$). Source data and exact *P* values are provided in the Source Data file.

repression[9]. Moreover, IκBζ is a critical downstream effector of EZH2-dependent chemokine and cytokine expression in psoriatic keratinocytes[35]. Thus, we hypothesized that IκBζ might repress *CXCL9*, *CXCL10*, and *CCL5* expression through the interaction with EZH2 and/or HDAC3 in melanoma. To test this hypothesis, control and IκBζ-overexpressing B16-F10 cells were treated with EZH2- or HDAC3-specific inhibitors, followed by gene expression analysis. As expected, overexpression of IκBζ significantly suppressed the expression of *Cxcl9*, *Cxcl10*, and *Ccl5* in B16-F10 cells. Notably, inhibition of EZH2 could restore gene expression of *Cxcl9* and *Cxcl10* in IκBζ-overexpressing and control B16-F10 cells, whereas HDAC3 inhibition fully reestablished *Cxcl10* and *Ccl5* expression in IκBζ-overexpressing B16-F10 cells (Fig. 9a, b).

In agreement with these findings, IκBζ did not only interact with HDAC3 and EZH2 in melanoma cells (Fig. 9c), but also led to increased recruitment of HDAC3 and potentially enhanced EZH2-dependent H3K27me3 levels at the promoter regions of *Cxcl9*, *Cxcl10*, and *Ccl5* (Fig. 9d). Conversely, knockdown of IκBζ in IκBζ-expressing LOX-IMVI cells diminished the enrichment for HDAC3 and H3K27me3 levels at the respective promoter sites (Fig. 9e). Importantly, we failed to detect binding of HDAC3 at the promoter regions of *IL1B* and *IL6*, confirming the specificity of HDAC3 recruitment to gene loci that are repressed by IκBζ (Supplementary Fig. S8c, d). Based on these findings, we propose that constitutively expressed IκBζ in melanoma recruits HDAC3 and EZH2 to the gene loci of *CXCL9*, *CXCL10*, and *CCL5*, leading to inaccessible promoter regions of these genes. As a consequence, cytotoxic T and NK cells are excluded from the TME, resulting in immunotherapy resistance.

## Discussion

IκBζ is a known co-factor of NF-κB, which has previously been found to activate a particular subset of NF-κB target genes, which predominantly encode cytokines, chemokines, and other immune-relevant proteins. While most previous studies focused on the role of IκBζ in immune cells, keratinocytes, or epithelial cells, its role in cancer remains largely unexplored[5,10,36]. Here we investigated the expression and function of IκBζ in melanoma. Interestingly, we found that a subgroup of melanoma patients and melanoma cell lines displayed constitutive protein expression of IκBζ, which did not correlate with its mRNA expression. These findings agree with a previous publication, which detected constitutive IκBζ expression in a subtype of diffuse large B-cell lymphoma, called ABC-DLBCL[11]. Mechanistically, it was shown that constitutive NF-κB activity in ABC-DLBCLs or over-activation of RAS signaling in keratinocytes can result in constitutive IκBζ expression[11,37]. However, we could rule out both pathways as drivers of constitutive IκBζ expression in melanoma, as neither the presence of activating *BRAF* or *NRAS* mutations nor increased NF-κB activity correlated with constitutive IκBζ protein expression. Instead, inhibition of the eIF4E/eIF4G cap-binding translation complex, but not the inhibition of transcription, efficiently suppressed IκBζ protein levels in IκBζ-expressing cells. Vice versa, inhibition of the proteasome restored IκBζ protein levels in non-expressing melanoma cell lines. Therefore, an altered activity or expression of the IκBζ-targeting E3

ubiquitin ligase PDLIM2[38], or known regulators of *NFKBIZ* mRNA stability, such as Regnase-1 and Regnase-3, could potentially account for the constitutive expression levels of IκBζ in melanoma[2,3,39]. Future studies on PDLIM2 or Regnase-1 and -3 might provide insights into the mechanism of constitutive IκBζ expression in melanoma cells.

As IκBζ mainly acts as a transcriptional co-regulator, we investigated target genes of IκBζ by RNA sequencing of IκBζ-deficient LOX-IMVI and D4M-3A cells. Interestingly, IκBζ predominantly regulated the expression of cytokines and chemokines in melanoma. Moreover, we could not only validate the direct binding of IκBζ to the respective target gene promoters but also revealed that overexpression of IκBζ alone was sufficient to induce target gene expression. Thus, we can rule out indirect signaling events that are needed to trigger IκBζ activation and its target genes in melanoma[40,41]. Our subsequent analysis revealed that IκBζ promoted the activation of the transcription factors STAT3 and NF-κB in melanoma, leading to increased gene expression of cytokines and growth factors, such as *IL1B* and *IL6*. As a result, we found that IκBζ expression promoted self-sustained tumor cell proliferation and tumor growth in an IL-6- and IL-1-dependent manner. These findings align with previous studies, showing that increased or constitutive expression of *IL1A*, *IL1B*, and *IL6* correlates with accelerated tumor growth and decreased overall survival in melanoma patients[42–45].

At the same time, IκBζ simultaneously repressed several other chemokines, known to regulate recruitment of T cells and NK cells into the tumor microenvironment, such as *CXCL9*, *CXCL10*, and *CCL5*[46]. Further analyses excluded STAT1/3 or NF-κB as effector molecules involved in IκBζ-mediated gene repression. Therefore, we investigated alternative transcriptional regulators associated with IκBζ signaling, such as EZH2 and HDACs[9,35]. Our experiments revealed that melanoma-derived IκBζ interacts with EZH2 and HDAC3, facilitating their recruitment to the respective promoter regions, resulting in silencing of gene expression. Consequently, inhibition of EZH2 or HDAC3 could re-establish chemokine expression in IκBζ-overexpressing melanoma cells, thereby validating HDAC3 and EZH2 as critical effectors downstream of IκBζ. In agreement with our results, recent publications unraveled EZH2 and HDAC3 as key mediators of immunotherapy resistance by repressing *Ccl5, Cxcl9*, and *Cxcl10*, leading to the exclusion of NK cells and cytotoxic T cells from the TME[29–34]. Previous reports also showed that CXCL10 expression in melanoma mediates the recruitment of cytotoxic T cells into the tumor stroma[47], whereas re-establishment of CCL5 expression induces NK-cell recruitment and melanoma regression[25]. Notably, we found that melanoma-derived IκBζ repressed cytotoxic T-cell or NK-cell recruitment in vivo, potentially impairing the effectiveness of immunotherapy by limiting tumor growth inhibition and cell death. Thus, our results support a model in which IκBζ-mediated recruitment of EZH2 and HDAC3 represses *Cxcl10, Cxcl9*, and *Ccl5* in melanoma, contributing to the observed lack of T-cell infiltration and immunotherapy resistance in the α-PD-1-treated, IκBζ-expressing melanoma mouse models. Accordingly, we propose that assessing IκBζ protein levels in tumor biopsies could serve as a diagnostic strategy to identify patients who would benefit from a combined immunotherapy with HDAC3 or EZH2 inhibitors.

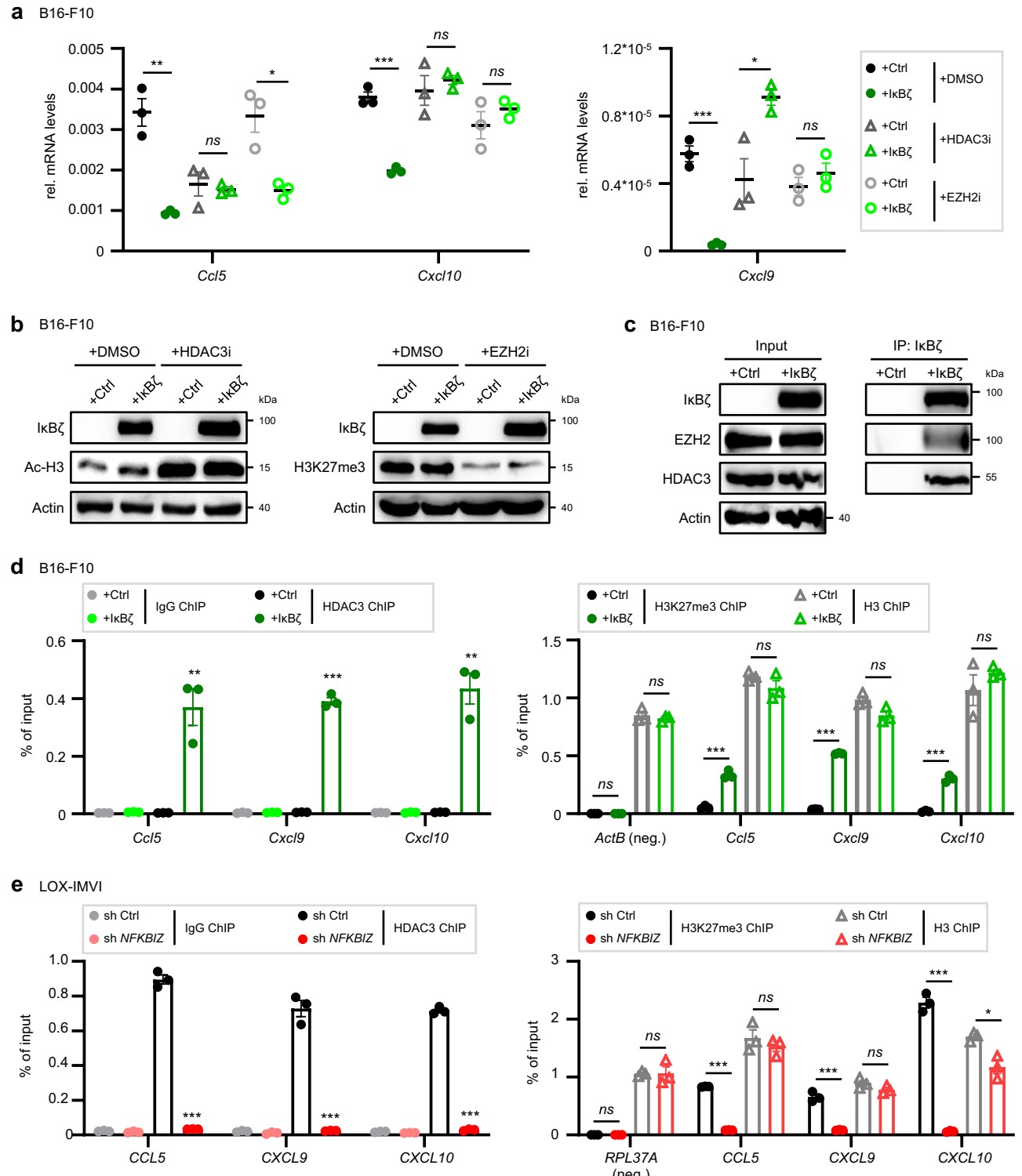

**Fig. 9 | IκBζ recruits EZH2 and HDAC3 to mediate the repression of *CCL5*, *CXCL9*, and *CXCL10* in melanoma cells. a** + **b** Control and IκBζ-overexpressing B16-F10 cells were treated for 24 h with 1.5 μM RGFP966 (HDAC3i) or 5 μM Taze-metostat (EZH2i). **a** Relative gene expression was normalized to *Actb*.
**b** Immunoblot analysis of IκBζ overexpression and inhibitor controls for HDAC3 inhibition (Ac-H3) and EZH2 inhibition (H3K27me3). β-Actin served as a loading control. **c** Co-immunoprecipitation (IP) of human IκBζ in control and IκBζ-overexpressing B16-F10 cells. Immunoblot detection was performed using an HDAC3 or EZH2 antibody. IκBζ detection served as a positive control for the co-immunoprecipitation. β-Actin was used as a loading control for the input. **d** *Left:* Chromatin immunoprecipitation (ChIP) assays of IgG (as control) and HDAC3 in

control and IκBζ-overexpressing B16-F10 cells. *Right:* ChIP assays of H3K27me3 and H3 in control and IκBζ-overexpressing B16-F10 cells. *Actb* served as a negative control for the H3K27me3 ChIP. **e** ChIP assays of IgG (as control) and HDAC3 in shRNA control and *NFKBIZ* knockdown LOX-IMVI cells. *Right:* ChIP assays of H3K27me3 and H3 in shRNA control and *NFKBIZ* knockdown LOX-IMVI cells. *RPL37A* served as a negative control for the H3K27me3 ChIP. Data derive from 3 independent experiments (mean ± standard deviation (*SD*)). Significance was calculated using a two-tailed Student's t-test (*p < 0.05, **p < 0.01, and ***p < 0.001, *ns* = not significant). Source data and exact *P* values are provided in the Source Data file.

Interestingly, we found a strong correlation between immunotherapy resistance, IκBζ protein expression, and lack of CD8[+] T-cell infiltration in melanoma patients. However, some individual patients who displayed constitutive IκBζ expression still responded to immunotherapy. As IκBζ cannot function alone, and aberrant expression of HDAC3 and EZH2 has already been described in melanoma[48,49], we speculate that a rare, functional deficiency or aberrant function of HDAC3 or EZH2 might account for the observed immunotherapy sensitivity in these IκBζ-expressing patients. Future mutation analysis or exon sequencing might help to understand why individual patients with IκBζ expression are still responsive to immunotherapy. This might facilitate the development of assays aimed at identifying patients who would benefit from immune checkpoint blockade. Of note, we also detected patients lacking tumor-derived IκBζ expression and immunotherapy resistance. We propose that this observation is not necessarily contradictory to our findings, as also other mechanisms might promote immunotherapy resistance in the absence of IκBζ. Importantly, when IκBζ protein expression was detectable, nearly all analyzed patients displayed T-cell exclusion from the TME and immunotherapy resistance. Thus, targeting IκBζ might constitute an attractive approach for sensitizing melanoma patients to immunotherapy, especially as IκBζ deletion would affect multiple oncogenic pathways, including NF-κB, STAT3, EZH2, and HDAC3.

Our results imply that IκBζ functionally interacts with STAT3 and NF-κB to induce the expression of multiple cytokines, including *IL1B* and *IL6*, whereas IκBζ can simultaneously interact with EZH2 and HDAC3 to repress the expression of *Ccl5, Cxcl9*, and *Cxcl10*. We propose that this dual function of IκBζ might be related to structural elements within the promoter region of its target genes that determine whether STAT3 or NF-κB can access and bind to the respective gene promoters. Both mechanisms of IκBζ are not necessarily mutually exclusive. Previous studies have demonstrated that EZH2 regulates STAT3 activity via methylation of the protein[50,51], while HDAC3 increases NF-κB activity through deacetylation of p65[52]. Consequently, the interaction of IκBζ with EZH2 and HDAC3 might be the most upstream event regulating IκBζ target genes in melanoma, whereas IκBζ-mediated gene activation derives from the additional presence of EZH2- and HDAC3-activated STAT3 and p65 at single gene promoters. Further research is needed to precisely explore the molecular mechanism of IκBζ-mediated gene activation and repression, particularly by further dissecting the crosstalk between EZH2, HDAC3, and IκBζ. Given that increased activity of STAT3, NF-κB, EZH2, and HDAC3 has also been reported in other tumors, it will be interesting to investigate the role of IκBζ in additional tumor entities.

## Methods

### Ethical statement
All animal experiments were approved by the local animal ethics committee (Landesuntersuchungsamt Rheinland-Pfalz; G22-01-012). All experiments involving patient material were approved by the local ethics committee of Mainz (Ethikkommission der Landesärztekammer Rheinland-Pfalz, No. 837.226.05(4884)) and Essen (Ethik-Kommission der Medizinischen Fakultät der Universität Duisburg-Essen, No. 11-4715). Informed consent was obtained from all participants. All patients agreed that the tumor material could be used for research purposes. Participants did not receive any compensation for their involvement in the study.

### Cell culture and treatment
LOX-IMVI, SK-MEL-30, and D4M-3A cells were maintained in RPMI-1640 medium, SK-MEL-3 cells in McCoy's 5 A medium, and YUMM1.7 cells in DMEM/F-12 medium, supplemented with 1% non-essential amino acids. All the other cell lines were maintained in DMEM medium. All media were supplemented with 10% FCS and antibiotics. Cells were grown at 37 °C in 5% $CO_2$. SK-MEL-2, SK-MEL-5, SK-MEL-28, Malme-3M,

UACC-62, and UACC-257 cells were obtained from the NCI-60 human tumor cell line panel. SK-MEL-3 (ACC321) and SK-MEL-30 (ACC151) cells were purchased from the Leibniz Institute DSMZ (Braunschweig, Germany). B16-F10 cells were ordered from ATCC (CRL-6475). LOX-IMVI, MV3, A375, MeWO, Mel2a, and HMB-2 cells were kindly gifted by Matthias Dobbelstein (University of Göttingen, Germany), D4M-3A cells were kindly provided by Tobias Sinnberg (University Hospital Tübingen, Germany). YUMM1.7 cells were kindly provided by Alpaslan Tasdogan (University Hospital Essen, Germany). Recombinant human IL-6 (Cat. 11340064) and IL-1β (Cat. 11340013) were purchased from Immunotools and used at a final concentration of 100 ng/mL. The following inhibitors were used: 4EGI-1 (Cayman, Cat. 15362), Actinomycin D (Merck, Cat. A9415), MG-132 (MedChemExpress, Cat. HY-13259), Stattic (Selleckchem, Cat. S7024), IMD-0354 (Selleckchem, Cat. S2864), Tazemetostat (Selleckchem, Cat. S7128), and RGFP966 (Cayman Chemical Cat. 16917).

### Lentiviral knockdown, overexpression, or CRISPR-Cas9-mediated knockout of melanoma cell lines
Knockdown or overexpression cells were generated as previously described[5]. In brief, lentiviral particles were produced in HEK293T cells using the lentiviral vectors pMD2.G (Addgene, Cat.12259) and pCMVR8.74 (Addgene, Cat.22036). For the generation of knockdown or overexpressing melanoma cells, the following constructs were used: pLKO.1 (sh Ctrl, Addgene, 8453), pLKO.1-TRCN0000147551 (sh*NFKBIZ*, Dharmacon), pLKO.1-TRCN0000020840 (sh*STAT3*, Addgene), pLKO-TRCN0000280021 (sh*STAT1*, Sigma), pLKO.1-TRCN0000014686 (sh*RELA*, Sigma) lenti-CRISPRv2 pRDI_292 and pRDI_292-NFKBIZ (all kindly provided by Stephan Hailfinger, Münster, Germany) and pAIOsgRNA mCMV-Nfkbiz (GSGM11941-247933929, Dharmacon). All melanoma cells were transduced in the presence of 8 μg/mL polybrene, followed by selection with 1-2 μg/mL puromycin (InvivoGen).

### Transient overexpression in melanoma cells
Transient overexpression was conducted using the LOX-IMVI Cell Avalanche® Transfection Reagent (EZ Biosystems, EZT-LOX-IMVI). The transfection reagent was mixed with Opti-MEM medium and the appropriate plasmids (pCR3.1 or pCR3-FLAG-NFKBIZ, both kindly provided by Stephan Hailfinger, Münster, Germany). After incubation for 15 min at RT, the DNA complexes were added dropwise to the cells and incubated for 5 h at 37 °C and 5% $CO_2$. The medium was then replaced by a fresh culture medium, and subsequent experiments were conducted 48 h post-transfection.

### Mice
All animal experiments were approved by the local animal ethics committee and administration (Landesuntersuchungsamt Rheinland-Pfalz, G22-1-012), and conducted in compliance with German laws and guidelines for animal care. Mice are housed under standard conditions at an average room temperature of 24 °C and a relative humidity of 50 ± 10%. A maximum of five animals are kept per cage. The light cycle follows a 12-h light/dark interval. Routine monitoring following the §11 permit is conducted daily by the animal care staff. For subcutaneous injection of the tumor cells into the back skin, 8-10-weeks 10-week-old female C57BL/6 mice were used (purchased from Charles River). 0.3*10^6 D4M-3A or YUMM1.7 control or *Nfkbiz* knockout cells, or 0.5*10^6 B16-F10 control or IκBζ-overexpressing cells were resuspended in 100 μL PBS and subcutaneously injected into the left and right back of the animals. For experiments involving immunodeficient mice, the same number of B16-F10 control and IκBζ-overexpressing cells were injected into 11-week-old female NOD/Scid mice (NOD.CB-17-Prkdc scid/Rj, Janvier Labs, France). All mouse experiments were performed with female mice to avoid high variation of immune cell subtypes due to gender differences. When the tumors became palpable, tumor

growth was assessed using a caliper, and the tumor volume was calculated using the following formula: tumor volume (mm3) = length (mm) x width (mm) x width/2 (mm). For treatment with α-PD-1 antibodies, mice received intraperitoneal injections with 200 µg α-IgG (Leinco Technologies, Cat.I-1177), or α-PD-1 antibodies (Leinco Technologies, Cat.P362), on days 7 and 10 (B16-F10 tumors), or days 7, 10, 13, and 16 (YUMM1.7 tumors), after injection of the tumor cells. Mice were sacrificed on day 11 (B16-F10), day 17 (YUMM1.7), or day 18 (D4M-3A) when at least one tumor reached a size of 1000 m3 or when the tumors started to ulcerate (termination criteria according to the animal study approval). The maximal tumor size or burden was not exceeded.

### Patient material, clinical data, and survival analysis

Tissue samples were obtained from the tissue biobanks of the University Medical Center Mainz and the West German Cancer Center Essen following their guidelines and approval of the local ethics committees (No. 837.226.05(4884) for Mainz and No. 11-4715 for Essen). Clinical characteristics of the patient cohort used to assess IκBζ levels in patients treated with ICB were stratified by clinical response criteria according to RECIST and are listed in Supplementary Data 4. Gender (as a biological attribute) was not considered in the collection or analysis of patient data, as melanoma affects females and males equally, and sex-based differences were not the focus of this study. Details on treatment and outcome specifics were recorded in an unidentifiable, pseudonymized format at the patient level. Biopsies for the evaluation of immunotherapy responses and their correlation with IκBζ expression were obtained before the start of the immunotherapy. To detect IκBζ in melanoma samples from different stages and anatomical sites, we used a commercial tissue microarray from BioCat GmbH (Cat. ME1002b-BX). Samples were categorized into three groups: no detectable IκBζ protein expression (graded as 0), moderate IκBζ protein expression (graded as 1), or high overall expression (graded as 2). IκBζ staining in infiltrating immune cells was neglected and excluded from the analysis. Subsequently, IκBζ levels were correlated with the clinical endpoints. The primary endpoints of this study were progression-free survival (PFS) and real-world tumor response following systemic therapy for metastatic stage IV upon adjuvant treatment failure. PFS was calculated from the start of initial treatment for metastatic stage IV until disease progression or death from any cause. Real-world tumor response was categorized as complete response (CR), partial response (PR), stable disease (SD), and progressive disease (PD) as described[53].

We employed Kaplan–Meier survival plots to illustrate median progression-free survival probabilities. Survival curves were compared using a log-rank test. In all cases, two-tailed $P$ values were calculated and considered significant for $p < 0.05$. Survival analyses were conducted using the survminer R package (RStudio Version 1.3.1093).

### Flow cytometry

To generate single-cell suspensions, tumors were chopped and digested for 30 min at 37 °C with shaking in 1 mL digestion mix (0.25 mg/mL Liberase TM (Sigma, 5401127001), 100 µg/mL DNase I (Sigma, 11284932001), and 0.5 mM CaCl$_2$ in RPMI-1640 medium without FCS). After digestion, the cell suspension was passed through a 100 µm cell strainer. Before the incubation with antibodies, samples were treated with Fc-Block (BioLegend, 101320) for 10 min at 4 °C and then surfaced-stained for 30 min at 4 °C with the following antibodies obtained from BioLegend: α-CD3-APC (Cat. 100236), α-CD4-PE (Cat. 100408), α-CD8-FITC (Cat. 100705), α-Ly6G-PE (Cat. 127608), α-Ly6C-APC (Cat. 128016), α-CD11c-PE (Cat. 117308), α-NK1.1-FITC (Cat. 156507), α-CD25-PE/Cy7 (Cat. 101915) and α-CD45-PE (Cat.103106). All antibodies were applied at a 1:50 dilution. DAPI (BioLegend, Cat.422801) was used to gate live cells. Data acquisition was performed at the LSRII flow cytometer (Becton Dickinson) with gate settings

based on the respective isotype controls. Analysis was performed using FlowJo software.

### Histology

After fixation in 4% formaldehyde overnight and paraffin-embedding, 5 µm sections were prepared. Antigen retrieval was performed in 1 mM EDTA (pH 8.0) for 40 min, followed by overnight incubation at 4 °C with the following antibodies: α-phospho-STAT3 (1:100 dilution, Cell Signaling, Cat. 9145), α-CD3 (1:100 dilution, Abcam, Cat. ab16669), α-CD8 (1:100 dilution Agilent, Cat. M7103) or a self-made rabbit antibody against human IκBζ (1:500, raised against the peptide CRKGADPSTRNLENEQ, ordered at Eurogentec). After incubation with a peroxidase-coupled secondary antibody, sections were stained with Vector NovaRED (Vector Laboratories, SK-4800) or the SignalStain® DAB Substrate Kit (Cell Signaling, Cat.8059), and counterstained with hematoxylin. For detection and quantification of PD-1L, paraffin-embedded tissue blocks were cut into 3 µm-thick sections, followed by deparaffinization and rehydration of the tissue using xylene and ethanol. Antigen retrieval was applied with Dako EnVision™ Flex Target Retrieval Solution pH 6.0 (Dako; Agilent Technologies, Inc.). As the primary antibody, a rabbit monoclonal α-PD-L1 (abcam; EPR19759 PUR; Fa. LOT: GR278006-15) was incubated at a 1:250 dilution in Dako EnVisionTM FLEX Antibody Diluent (Dako; Agilent Technologies, Inc.) in the Thermofisher Autostainer 480-B (Thermo Scientific). As secondary antibodies, tissues were incubated with EnVision™ Flex+Rabbit Linker (Dako; Agilent Technologies, Inc.). For color reaction, the Dako EnVision™ Flex Substrate buffer (Dako; Agilent Technologies, Inc.) and Dako EnVision™ Flex DAB+ Chromogen (Agilent Technologies, Inc.) were applied. Finally, the tissues were counterstained using hematoxylin (Dako; Agilent Technologies, Inc.). The slides were digitized with a NanoZoomer 2.0-HT scanner (Hamamatsu Photonics, Herrsching, Germany). Pictures were captured using the NDP.View2 (Hamamatsu) software. For quantification of PD-L1 staining, the clinically well-established and broadly used Tumor Proportion Score (TPS) was used, which quantifies linear membrane staining exclusively on tumor cells.

### Detection of NFKBIZ mRNA levels via RNAScope

The RNAScope™ 2.5 HD Red Assay (Advanced Cell Diagnostics, ACD, 322360) was performed following the manufacturer's protocol. The RNAscope™ Probe Hs-NFKBIZ (ACD, 497851) was used to detect NFKBIZ mRNA expression. The protocol was slightly adapted: Amp5 incubation was extended to 40 min, Amp6 incubation to 20 min, and the detection step using RED reagents was performed for 15 min. Images were acquired using the BX53 microscope from Olympus (40× magnification objective). For quantification, the number of red punctate signals per cell was counted in 100 cells per image, with three images (optical windows) analyzed per sample. The mean signal per cell was first determined for each individual image, and subsequently, the average of the three images was calculated to derive a single representative value for NFKBIZ mRNA expression per sample.

### Immunofluorescence analysis

Tumor tissue was fixed overnight at 4 °C with 4% paraformaldehyde, followed by incubation in 15% sucrose for 24 h and 30% sucrose for another 24 h. Tissue was embedded in Tissue-Tek® O.C.T.™ Compound, and 6 µm cryosections were cut. Slides were rinsed for 2 min with PBS. Tissue was permeabilized with 0,05% Triton X-100 in PBS for 10 min at 37 °C, followed by blocking with 4% normal donkey serum (NDS) in 0.05 %Triton-X-100 for 1 h at RT. Slides were rinsed with PBS for 3 × 5 min followed by incubation with α-CD45-FITC (Biolegend #103108) and α-CXCL9-AF647 (Biolegend #515606) in 2 % NDS at 4 °C overnight. The next day, the tissue was rinsed for 3 × 5 min with PBS and covered with Fluroshield + DAPI. Sections were dried at RT for 2 h and stored at 4 °C, protected from light. Image acquisition was performed using a Keyence BZ-X microscope with DAPI, GFP, and Cy5 filters.

 

## Cell viability assay

To assess cell viability and proliferation, 1000 – 2000 melanoma cells were seeded on a white 96-well plate per well (PerkinElmer, Cat.6005680), 24 h before the first measurement. In some experiments, control or IκBζ-overexpressing MV3 cells were stimulated with 100 ng/mL of IL-6 or IL-1β, 24 h before the first measurement. For treatments with supernatant (SUP), IκBζ-overexpressing MV3 cells were incubated in DMEM medium without FCS for 48 h, after which supernatants were collected. After centrifugation, MV3 control cells were resuspended in the supernatant and seeded for the cell viability assay. During the viability measurement, cells were incubated at 37 °C and 5% $CO_2$. The CellTiter-Glo® Luminescent Cell Viability Assay (Promega, Cat.G7571) was performed according to the manufacturer's instructions. The relative luminescence was detected using the Hidex Sense Microplate Reader. Relative luminescence units (RLU) retrieved from samples 24 h after seeding were set to 100%, and the relative cell growth was assessed by calculating the fold change in RLUs compared to the 24-h-post-seeding value.

## Immunoblot analysis and co-immunoprecipitation

Immunoblot analysis and co-immunoprecipitation were performed as described[35]. The following antibodies from Cell Signaling were used for immunoblot analysis: α-IκBζ (Cat. 9244), α-phospho-IκBα (Cat. 2859), α-IκBα (Cat. 9242), α-phospho-STAT1 (Y701, Cat. 9167), α-STAT1 (Cat. 14994), α-phospho-STAT3 (Y705, Cat. 9145), α-STAT3 (Cat. 4904), α-NF-κB1 p105/p50 (Cat. 3035), anti-NF-κB p65 (Cat. 8242), α-EZH2 (Cat. 5246), α-Tri-Methyl-Histone H3 (Lys27) (Cat. 9733), α-HDAC3 (Cat. 85057) and α-β-Actin (Cat. 3700). The α-acetyl-Histone-H3 antibody (Merck, Cat. 06-599) was kindly provided by Oliver Krämer (University Medical Center Mainz, Germany). For the detection of mouse IκBζ, a self-made antibody was used[54]. All phospho-specific antibodies were used at a 1:500 dilution. β-Actin, GAPDH, and histone H3 antibodies were used at a 1:5000 dilution. All other antibodies were used at a 1:1000 dilution. For co-immunoprecipitation of human IκBζ, HEK293T cells were transiently transfected with a pCR3-FLAG-NFKBIZ construct. After cell harvesting, protein lysates were pre-cleared with protein A/G PLUS agarose beads (Santa Cruz, sc-2003) for 1 h at 4 °C. The pre-cleared lysates were then incubated overnight at 4 °C with antibodies against IκBζ (Cell Signaling, Cat. 9244; Novus Biologicals, Cat. NBP1-89835, or a custom-made antibody from Eurogentec raised against the IκBζ peptide CRKGADPSTRNLENEQ), or against β-Gal (Santa Cruz, Cat. sc-19119), or IgG (Abcam, Cat. ab46540) as control. For each co-immunoprecipitation, 5 μL of antibody was used. The immune complexes were precipitated with protein A/G PLUS agarose beads for 2 h at 4 °C, washed with Co-IP buffer, and eluted using 6x SDS-PAGE sample buffer. Pulldown samples were analyzed by SDS-PAGE and detected by immunoblotting using a FLAG antibody (Sigma, Cat. F1804), HDAC3 (Cell Signaling, Cat. 85057), or EZH2 antibody (Cell Signaling, Cat. 5246). Antibodies for immunodetection were used at a 1:1000 dilution.

## Chromatin fractionation and chromatin immunoprecipitation

Chromatin fractionation was performed as published before[55]. α-GAPDH (Cell Signaling, Cat. 2118) and α-histone H3 antibodies (Abcam, Cat. ab1791) were used as controls for the chromatin-unbound and chromatin-bound fractions, respectively. ChIP assays were performed as described[35]. In brief, chromatin was prepared by crosslinking with 0.25 M DSG (Thermo Fisher, Cat. 20593) for 45 min at RT, followed by cross-linking with 1 % formaldehyde (Thermo Fisher, Cat. 28906). After 20 min sonication, chromatin was incubated with protein G-coupled Dynabeads (Thermo Fisher, Cat.10003D) and 2 μg self-made antibody against IκBζ (raised against the peptide CRKGADPSTRNLENEQ, ordered at Eurogentec), 3 μg α-STAT3 (Thermo Fisher, Cat. MA1-13042), 2 μg α-NF-κB p65 (Diagenode, Cat. C15310256), 5 μL α-NF-κB1 p105/p50 (Cell Signaling, Cat. 3035), 5 μL α-HDAC3 (Cell Signaling, Cat.

85057), 2 μg α-Tri-Methyl-Histone H3 (Lys27) for ChIP assays in LOX-IMVI cells (Diagenode, Cat. C15200181), 5 μL α-Tri-Methyl-Histone H3 (Lys27) (Cell Signaling, Cat. 9733) for ChIP assays in B16-F10 cells, 1 μg α-Histone H3 (Abcam, Cat. ab1791) or an IgG antibody as control (Abcam, Cat. ab46540), overnight at 4 °C on a rotator. ChIP primers for IκBζ target genes are listed in Supplementary Data 3. The primers were self-designed for the promoter regions corresponding to transcription factor-bound sites of STAT3 and p65. Maxima SYBR Green Master Mix (Thermo Fisher, Cat. K0221) was used to perform quantitative PCR. The percentage of input was calculated as described[35].

## Gene expression analysis by qPCR and bulk RNA sequencing

The procedure of total RNA extraction and quantitative PCR has been published before[5]. In brief, total RNA was extracted using Qiazol (Qiagen, Cat. 79306), followed by DNase I digestion to remove contaminating DNA. Afterwards, RNA was either subjected to cDNA synthesis (for qPCR analysis) or library preparation followed by bulk RNA sequencing. Real-time PCR analysis was performed using the CFX384 Touch Real-Time PCR System (Bio-Rad) under the following PCR conditions: initial denaturation 15 min at 95 °C, followed by 40 cycles of 95 °C for 15 seconds and 60 °C for 45 seconds. Relative gene expression was quantified by real-time PCR using a Green master mix (Genaxxon, Cat. M3023) and self-designed primers (Supplementary Data 2). For the calculation of the relative mRNA levels, the target genes' Ct values were normalized to the Ct values of the reference genes *RPL37A* for human samples and *Actb* or *Hprt1* for murine samples using the $2^{-\Delta CT}$ method.

For transcriptome analysis, both sequencing and library construction from total RNA were performed by Novogene (Munich, Germany) in the case of D4M-3A cells, or by the Genomic Core Facility (Münster, Germany) in the case of LOX-IMVI cells. For RNAseq analysis of the human samples, libraries were constructed with the Ultra RNA Library Prep Kit, and sequencing was performed using the Illumina NextSeq High Output kit. For analysis of the D4M-3A cells, quantified libraries were sequenced on the Illumina Novaseq 6000 platform with paired-end 150 cycle (PE150) sequencing, according to effective library concentration and data amount. The sequencing strategy was paired-end 150 cycles (PE150). CLC Genomics workbench (v24.0; Qiagen) was used to further process the sequencing raw reads. Mapping against the human reference genome hg38 (GRCh38.109) or the mouse reference genome mg39 (GRCm39) was performed by the NGS core facility (Research Center for Immunotherapy) using CLC's default settings (mismatch cost = 2; insertion cost = 3; deletion cost = 3; length fraction = 0,8; similarity fraction = 0,8; global alignment = No; strand-specific = both; library type = bulk; maximum number of hits for a read = 10; count paired reads as two = no; ignore broken pairs = yes). Differentially expressed genes were filtered for a minimum absolute fold change > 2 with a difference cut-off of absolute > 4 and an adjusted $P$ value ≤ 0.05, between control and *NFKBIZ* knockdown or knockout cells. Raw data of the gene expression data sets were published under the GEO accession numbers GSE277887 and GSE277888. Heatmaps were generated using the Morpheus software (Morpheus, https://software.broadinstitute.org/morpheus). Conserved IκBζ target genes deriving from LOX-IMVI and D4M-3A cells, and genes that are only expressed in one species (human or mouse), are presented in Supplementary Data 1.

## Statistics & reproducibility

For all experiments, at least 3 independent biological replicates were generated and analyzed. Subsequently, significance was calculated using a two-tailed Student's t-test (*$p < 0.05$, **$p < 0.01$, and ***$p < 0.001$). For all cell culture experiments, the standard deviation is displayed. The standard error is shown for all the data derived from mouse experiments. For all mouse experiments, sample size was calculated using a t-test (one-tailed, effect size 2, alpha = 0.05 and beta = 0.2). For the analysis of the patient samples, the sample size was

not pre-determined. Instead, all available patient samples with detailed clinical data, deriving from two different locations, were used. No data were excluded from the analyses. Analysis and evaluation of the patient samples were performed blindly by two independent researchers. All mice were randomly allocated to the treatment groups.

## Reporting summary

Further information on research design is available in the Nature Portfolio Reporting Summary linked to this article.

## Data availability

Raw data from the bulk RNA sequencing generated in this study have been deposited at the Gene Expression Omnibus (GEO) repository of the National Center for Biotechnology Information (NCBI) under the accession numbers GSE277887 and GSE277888. Patient data containing potentially identifying information are protected by privacy laws and are not generally available to the public. However, anonymized datasets supporting the findings of this study are provided in the Source Data file. Source Data are provided with this paper. Additional and raw data generated in this study are provided in the Supplementary Information and the Source Data file. Source data are provided with this paper.

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

## Acknowledgements

The study was supported by grants from the Peter Hans Hofschneider professorship of Molecular Medicine, funded by the Foundation for experimental biomedicine (D.K. and A. Tasdogan (hereinafter referred to as A.T.), the Deutsche Forschungsgemeinschaft TRR156/3 (project number 246807620, B09 and B11) (D.K. and S.G.), TRR355/1 (project number 490846870, A08) (D.K.), SFB 1292/2 (project number 318346496, TPQ1, TP22, TP08) (M.M.G and C.D.), the DFG Emmy Noether Award (DFG, 467788900) (A.T.), and the DFG Emmy Noether Program (DE 2654/2-1) (C.D.). Furthermore, D.K. is supported by the Rise up! program of the Boehringer Ingelheim Foundation (BIS), C.D. is supported by the Federal Ministry for Education and Research (BMBF), grant number 03ZU1202GA, and A.T. is supported by the Ministry of Culture and Science of the State of North Rhine-Westphalia (NRW-Nachwuchsgruppenprogramm), and an ERC Starting grant (METATARGET, 101078355). We thank Claudia Braun from the Histology Core Facility (University Medical Center of the Johannes Gutenberg-University Mainz), Bonny Adami, Bettina Mros, and Dagmar Löck (University Medical Center of the Johannes Gutenberg-University Mainz) for excellent experimental support. Human biological samples and related data were provided by the Westdeutsche Biobank Essen (WBE/SCABIO, University Hospital Essen, University of Duisburg-Essen, Essen, Germany; approval no. 11-4715) and the Tissue Biobank of the University Medical Center Mainz, approval no. 837.226.05(4884). The Schemes in Fig. 6 (Kramer, D. (https://BioRender.com/e11y470) and 7 (Kramer, D. (https://BioRender.com/k58w881) were created in BioRender.

## Author contributions

A.K., A.-M.K.-M., A. Stastny, V.V., and C.D. performed experiments and data analysis. M.K. analyzed the RNA sequencing data. T.S. provided D4M-3A cells, K.S.-O. provided cells from the NCI 60 panel, and T.S. and A. Tasdogan (hereinafter referred to as A.T.) helped to plan the mouse experiments. B.W.-B., H.S., M.H., A. Tuettenberg, A. T., G.A., A. Sucker, L.J.A., D.S., and S.G. selected the patient samples and helped in interpreting the human data. M.M.G. provided patient material and performed some histological stainings and clinical evaluation of the data. A.K. and D.K. drafted the first version of the manuscript. All authors reviewed, provided feedback, and approved the final manuscript.

## Funding

## Competing interests

The authors declare no competing interests.

## Additional information

[1]Department of Dermatology, University Medical Center of the Johannes Gutenberg University of Mainz, Mainz, Germany. [2]Institute for Immunology, University Medical Center of the Johannes Gutenberg University Mainz, Mainz, Germany. [3]Research Center for Immunotherapy, University Medical Center of the Johannes Gutenberg University of Mainz, Mainz, Germany. [4]Department of Microbiology & Immunology, Stanford University School of Medicine, Stanford, CA, USA. [5]German Cancer Consortium (Deutsches Konsortium für Translationale Krebsforschung) and German Cancer Research Center (Deutsches Krebsforschungszentrum), Heidelberg, Germany. [6]Department of Molecular Medicine, Interfaculty Institute of Biochemistry, University of Tübingen, Tübingen, Germany. [7]Cluster of Excellence iFIT (EXC 2180) "Image Guided and Functionally Instructed Tumor Therapies", Eberhard Karls University Hospital Tübingen, Tübingen, Germany. [8]Department of Dermatology, Eberhard Karls University Hospital Tübingen, Tübingen, Germany. [9]Department of Dermatology, Venereology and Allergology, Charité - University Medical Center Berlin, Berlin, Germany. [10]Department of Dermatology, Venereology and Allergology, University Hospital Essen, NCT West, Campus Essen, German Cancer Consortium, Partner Site Essen (DKTK) & University Alliance Ruhr, Research Center One Health, Essen, Germany. [11]Institute of Pathology, University Medical Center of the Johannes Gutenberg University Mainz, Mainz, Germany. [12]TRON, Translational Oncology at the University Medical Center of the Johannes Gutenberg University Mainz, Mainz, Germany. [13]Center for Thrombosis and Hemostasis, University Medical Center of the Johannes Gutenberg University Mainz, Mainz, Germany. ✉e-mail: kramerda@uni-mainz.de

