## [Transparent Peer Review file · Nature Communications]

Constitutive expression of the transcriptional co-activator I κ B ζ promotes melanoma growth and immunotherapy resistance

Corresponding Author: Professor Daniela Kramer

Version 0:

Reviewer comments:

Reviewer #1

(Remarks to the Author)

In this study, the authors investigate a novel role of the NF- κ B co-regulator I κ B ζ in solid tumors. Although I κ B ζ expression varies across tumor types, they found that it is constitutively expressed in over 30% of cases. They noticed that I κ B ζ expression is mainly regulated at the post-translational level. Inhibition of constitutive I κ B ζ expression suppresses NF- κ B and STAT3(also STAT1) chromatin binding and reduces IL-1 β and IL-6 expression. Constitutive I κ B ζ expression appears to inhibit specific chemokines, resulting in impaired recruitment of NK cells and CD8+ T cells to tumors and resistance to α -PD-1 immunotherapy in mouse models. I κ B ζ expression was also associated with the absence of CD8+ T cells in humans and disease progression during immunotherapy.

These findings reveal a new regulatory mechanism of I κ B ζ in cancer beyond its known functions in immune cells, highlighting a unique role for this non-enzymatic protein.

The experimental design is highly robust, effectively addressing the heterogeneous expression patterns of I κ B ζ . While further research could explore variations in constitutive I κ B ζ expression levels, the overall study is compelling.

Comments:

- Figures 4F and 5G are the same experiment, but the actin image is the same. It should be improved by emphasizing that they are the same experiment or by replacing the image to avoid misunderstanding.
- In Figure 4F, the text suggests that I κ B ζ expression does not change with α -PD-1 treatment and that pSTAT3 is increased, but this is not clearly conveyed. A repeat experiment is recommended.
- The legend suggests that Figure 5F involves I κ B ζ depletion and Figure 5G a proliferation assay, but data for these are missing.
- The FACS plot in Figure 3J is too small.
- In D4M-3A cells, the CRISPR knockout of I κ B ζ appears to retain more I κ B ζ protein compared to shRNA knockdown (e.g., Figures 5A, S3D, S5A). Could this be due to the proliferation of non-KO cell populations over time in culture?
- Is the homemade I κ B ζ antibody only for IHC?

Reviewer #2

(Remarks to the Author)

Kolb and colleagues show in this manuscript that the co-regulator of NF- κ B, I κ B ζ , can promote melanoma cell growth and contribute to T cell exclusion in the tumor microenvironment (TME), impairing the response to immune checkpoint blockade by anti-PD-1. They found that I κ B ζ is highly expressed in around one third of melanomas and is associated with worst outcomes upon ICB, which could help to stratify patients under this treatment if validated in additional cohorts.

Mechanistically, the authors demonstrate that I κ B ζ regulates the expression of multiple inflammatory genes by promoting the activation and chromatin binding of STAT1, STAT3 and p65. They validated the induction of IL1B and IL6 expression by I κ B ζ together with the downregulation of CCL5, CXCL9 and CXCL10 in multiple cell lines (both human and murine), which led to increased tumor growth and the exclusion of NK and T cells in the TME. Although the regulation of these genes by I κ B ζ was

previously described in keratinocytes, epithelial and immune cells (such as macrophages), this is the first report of this factor acting intrinsically in melanoma cells, revealing a potential therapeutic target in combination with ICB.

However, there are some aspects that were not addressed in this manuscript and would help to strengthen the author's conclusions:

Main points

1. The authors show the correlation of I κ B ζ expression with the response and progression free survival of patients treated with ICB (figs 1 and 4), but they don't indicate whether the samples analyzed were collected before or after treatment. I κ B ζ could be overexpressed as an adaptive resistance mechanism to ICB and, although ICB didn't affect I κ B ζ expression in B16 tumors, it should be tested in patients. Some biomarkers have shown stronger correlations early on-treatment or after treatment than at baseline, for example the induction of inflammatory genes and T cell infiltration (PMID: 27301722, etc). The expression of I κ B ζ target genes could be analyzed in pre- and post-treatment samples from published data sets, addressing if their up- and downregulation correlate with patient outcomes.

Authors should also test whether I κ B ζ expression is an independent or better predictor than other correlates of response, such as increased T cell infiltrates and immune signatures previously described. Co-expression of I κ B ζ and STAT3-P or p65 could also improve the predictive potential.

2. The impact of I κ B ζ on ICB response is addressed by its overexpression in B16-F10 cells treated with anti-PD-1. However, to demonstrate its potential as therapeutic target, I κ B ζ downregulation in anti-PD-1 resistant cells (e.g. YUMM1.7) would be more relevant.

It would be interesting as well to address the changes in the TME driven by I κ B ζ blockade together with anti-PD-1.

Minor

1. Authors do not justify why p50 is not included in the mechanistic assays when it is one of the known transcription factors that can interact with I κ B ζ together with p65 or STAT3.

2. The RNAseq analysis of I κ B ζ KO cells showed little overlap of differentially expressed genes between the 2 cell lines studied (LOX-IMVI and D4M-3A). Indeed, only one gene in common is related with immune pathways, which are the functions of other 18 genes in common? All of that should be discussed.

Reviewer #3

(Remarks to the Author)

The manuscript from Kolb et al reports interesting results surrounding a role for I κ B ζ (hereafter IKBz) in melanoma. As the authors note, IKBz expression has not been studied extensively in solid tumors, so the results are important in order to better understand the oncogenic functions study this enigmatic member of the I κ B family. The authors utilize melanoma cell lines, corresponding cell line-derived tumor studies, and analysis of patient of samples to approach oncogenic mechanisms associated with IKBz. Key reported data are: IKBz protein levels are not correlated with mRNA levels, about 35% of melanoma tumors express IKBz, IKBz expression predicts poor outcomes, and IKBz drives tumor growth and suppression of T cell recruitment and correspondingly resistance to immunotherapy. IKBz modulates recruitment of STAT3 and NF- κ B RelA to some promoters. For some tumors, there is a correlation between IKBz expression and response to checkpoint inhibitors. While there are clearly some interesting results, some concepts are not well developed and remain unclear.

1) While cell line data do not show a correlation between IKBz mRNA levels and protein, the authors should analyze TCGA melanoma datasets for IKBz mRNA levels along with proposed target genes. Maybe human tumors show a correlation between IKBz mRNA and protein levels and cell lines are not a good model for this analysis.

2) Fig. 1 reports that MG132 stabilizes IKBz protein levels, but for one of the cell lines MG132 increases the mRNA and for another that is only a weak increase in protein levels. It was unclear the interpretation of the effect of the cap-dependent translation inhibitor.

3) In Fig. 1, how were the Kaplan-Meier results obtained? There was nothing in the main text about how the data were derived.

4) Gene expression analysis using cell lines and knockdown or overexpression of IKBz revealed previously identified targets such as IL-1b and IL-6. It was unclear relative to other targets as to their common regulation across the cell lines. Additionally, genes upregulated by IKBz knockdown were not a focus of this section (and come up later). For example, Cxcl9, Cxcl10 – what were the effects on IKBz knockdown/overexpression among the cell lines? And see comment above regarding TCGA analysis to correlate with the cell line data. Generally consistent with these results, IKBz ChIPs at some of these promoters, including Cxcl10 (and see below).

5) IKBz was shown to promote cell growth/viability (although the relative contribution of either mechanism is not clear). IKBz expression in B16 cells/tumors promotes tumor growth in C57/BL6 mice. In this regard, the authors should do the same study in immune-deficient mice to determine the impact of the endogenous immune system on tumor growth which was not addressed specifically, although implied.

6) IKBz expression suppressed recruitment of CD8 T cells into the tumors, and IKBz-expressing tumors showed reduced expression of Cxcl9, and 10. The question is whether the loss of Cxcl9 and 10 is intrinsic to the tumor and its level of IKBz expression or due to lack of TME-derived chemokines in the tumor microenvironment (see point above pt. 4 above, and point 8 and 9 below).

7) Human patient data suggest an inverse correlation between IKBz expression and response to immunotherapy for a portion of patients. Notably, there are both responders and non-responders among those patients with tumors that don't express IKBz, so that additional mechanisms must be at play to promote poor response to therapy.

8) IKBz expression in the B16 tumors suppressed the response to anti-PD-1. In these studies, anti-PD-1 promoted recruitment of T cells into the tumor and this was blocked with IKBz tumor expression. Interestingly, it was the anti-PD-1 treatment that leads to an increase of Cxcl9 and 10, and this was blocked by IKBz expression. Regarding points above about repression of Cxcl9 and 10, are they suppressed in the cell lines or is it that in tumors, IKBz suppresses T cell recruitment

which leads to lack of chemokines to induce their expression?

9) There is no explanation for the effects of IKBz on STAT1/3 phosphorylation. Does IKBz regulate JAK/STAT activity? Regarding the effect of IKBz on repression of Cxcl9 and 10 – does IKBz promoter localization block recruitment of STAT proteins and NF-kB to the promoter? Thus there is no explanation for how IKBz suppresses Cxcl9 and 10 gene expression (and see points above).

10) Does IKBz colP with RelA and STAT3? I see the experiment described in the Methods, but don't see the description in the main text.

Version 1:

Reviewer comments:

Reviewer #1

(Remarks to the Author)

The authors have appropriately addressed my comments.

Reviewer #2

(Remarks to the Author)

The authors addressed all main issues highlighted by the reviewers and the revised manuscript has substantially improved. Extensive additional data was provided, and the conclusions are strongly supported.

Reviewer #3

(Remarks to the Author)

The authors have addressed my concerns in a thorough manner. The work provides important insight into mechanisms associated with immunotherapy resistant melanoma, and also provide new insight into mechanisms associated with I κ Bzeta.

Prof. Dr. rer. nat. Daniela Kramer
Group leader
Geb. 401 / EG / Raum 0.016
Langenbeckstr. 1
55131 Mainz
Telefon: +49 (0) 6131 17-5731
Telefax: +49 (0) 6131 17-472917
E-Mail: kramerda@uni-mainz.de
www.hautklinik-mainz.de

Mainz, 01.04.2025

Point-by-point response to the Reviewers' comments

Reviewer #1 (Remarks to the Author); expert in immunology:

In this study, the authors investigate a novel role of the NF- κ B co-regulator I κ B ζ in solid tumors. Although I κ B ζ expression varies across tumor types, they found that it is constitutively expressed in over 30% of cases. They noticed that I κ B ζ expression is mainly regulated at the post-translational level. Inhibition of constitutive I κ B ζ expression suppresses NF- κ B and STAT3 (also STAT1) chromatin binding and reduces IL-1 β and IL-6 expression. Constitutive I κ B ζ expression appears to inhibit specific chemokines, resulting in impaired recruitment of NK cells and CD8⁺ T cells to tumors and resistance to α -PD-1 immunotherapy in mouse models. I κ B ζ expression was also associated with the absence of CD8⁺ T cells in humans and disease progression during immunotherapy. These findings reveal a new regulatory mechanism of I κ B ζ in cancer beyond its known functions in immune cells, highlighting a unique role for this non-enzymatic protein. The experimental design is highly robust, effectively addressing the heterogeneous expression patterns of I κ B ζ . While further research could explore variations in constitutive I κ B ζ expression levels, the overall study is compelling.

We thank the reviewer for the positive comments.

Comments:

- Figures 4F and 5G are the same experiment, but the actin image is the same. It should be improved by emphasizing that they are the same experiment or by replacing the image to avoid misunderstanding.
- In Figure 4F, the text suggests that I κ B ζ expression does not change with α -PD-1 treatment and that pSTAT3 is increased, but this is not clearly conveyed. A repeat experiment is recommended.

As suggested, we completely repeated the immunoblot analysis of I κ B ζ from tumor cell lysates. The data can be now found in Figure 6d. We omitted the immunoblot analysis of pSTAT3 from the tumor cell lysates (originally Figure 5G), as we could largely rule out a direct STAT3-dependent regulation of I κ B ζ -repressed chemokines (see also Supplementary Figure S8). Consequently, this analysis in α -PD-1-treated tumors is not considered informative.

- The legend suggests that Figure 5F involves IκBζ depletion and Figure 5G a proliferation assay, but data for these are missing.

We apologize for this error. The legends for Figure 5f and Figure 5g have now been corrected. Due to some data rearrangements, Figure 5F was moved to the Supplementary data and is now presented as Supplementary Figure S7f. It refers to a tumor cell viability assay using an IKK or STAT1/3 inhibitor to treat control and IκBζ-overexpressing MV3 cells. As already pointed out, we removed Figure 5G from the manuscript.

- The FACS plot in Figure 3J is too small.

We increased the overall size of the FACS plots. Due to figure rearrangements, Figure 3J is now presented as Figure 4h.

- In D4M-3A cells, the CRISPR knockout of IκBζ appears to retain more IκBζ protein compared to shRNA knockdown (e.g., Figures 5A, S3D, S5A). Could this be due to the proliferation of non-KO cell populations over time in culture?

We did not isolate single-cell clones following the CRISPR-Cas9 knockout; we only selected cells based on their puromycin resistance. Therefore, it is possible that individual cells retaining *Nfkbiz* expression may overgrow the *Nfkbiz* knockout cells over time. We had difficulties keeping *Nfkbiz* KO D4M-3A cells in culture for a longer time without losing the knockout (data not shown), suggesting that cells without *Nfkbiz* deletion could potentially overgrow the culture over time. We did not follow up on this issue further.

- Is the homemade IκBζ antibody only for IHC?

The homemade antibody is suitable for the detection of IκBζ protein in its native conformation, but not under denaturing conditions such as SDS-PAGE. We thoroughly validated the antibody by immunostaining human IκBζ-overexpressing B16-F10 cells, and by co-immunoprecipitation experiments, comparing the antibody to commercially available IκBζ antibodies (Supplementary Figure S1d and S1e). In both experiments, the antibody was able to specifically detect IκBζ. Moreover, also in chromatin immunoprecipitation (ChIP) assays, we could specifically detect IκBζ binding to gene promoters in control but not in *NFKBIZ* knockdown LOX-IVMI cells using the self-made IκBζ antibody (Figure 2h). However, the antibody failed to detect IκBζ in Western blot experiments using control and *NFKBIZ* knockdown LOX-IVMI cells (data not shown).

Reviewer #2 (Remarks to the Author); expert in melanoma and immunotherapy:

Kolb and colleagues show in this manuscript that the co-regulator of NF-κB, IκBζ, can promote melanoma cell growth and contribute to T cell exclusion in the tumor microenvironment (TME), impairing the response to immune checkpoint blockade by anti-PD-1. They found that IκBζ is highly expressed in around one-third of melanomas and is associated with worst outcomes upon ICB, which could help to stratify patients under this treatment if validated in additional cohorts. Mechanistically, the authors demonstrate that IκBζ regulates the expression of multiple inflammatory genes by promoting the activation and chromatin binding of STAT1, STAT3, and p65. They validated the induction of IL1B and IL6 expression by IκBζ together with the downregulation of CCL5, CXCL9, and CXCL10 in multiple cell lines (both human and murine), which led to

increased tumor growth and the exclusion of NK and T cells in the TME. Although the regulation of these genes by I κ B ζ was previously described in keratinocytes, epithelial, and immune cells (such as macrophages), this is the first report of this factor acting intrinsically in melanoma cells, revealing a potential therapeutic target in combination with ICB. However, there are some aspects that were not addressed in this manuscript and would help to strengthen the author's conclusions:

We thank the reviewer for the important comments on our work. As requested, we added more data to the manuscript to strengthen our conclusions.

Main points

1. The authors show the correlation of I κ B ζ expression with the response and progression-free survival of patients treated with ICB (figs 1 and 4), but they don't indicate whether the samples analyzed were collected before or after treatment.

The human melanoma samples used in Figure 1 and Figure 4 were derived from biopsies taken before the start of immunotherapy. We added this information to the Material and Methods section. Moreover, we performed I κ B ζ staining of biopsies taken pre- and post-immunotherapy from the same patient. This data is now added in Supplementary Figure S5c.

I κ B ζ could be overexpressed as an adaptive resistance mechanism to ICB and, although ICB didn't affect I κ B ζ expression in B16 tumors, it should be tested in patients. Some biomarkers have shown stronger correlations early on-treatment or after treatment than at baseline, for example the induction of inflammatory genes and T cell infiltration (PMID: 27301722, etc). The expression of I κ B ζ target genes could be analyzed in pre- and post-treatment samples from published data sets, addressing if their up- and downregulation correlate with patient outcomes.

We thank the reviewer for this very important comment. While we agree that investigation of I κ B ζ expression in response to ICB is critical, this issue has several challenges. Specifically, it might be difficult to discriminate between the cause and consequence, i.e. whether ICB modulates I κ B ζ expression or I κ B ζ expression modulates the response to ICB.

To explore this issue, we analyzed I κ B ζ protein expression in human melanoma samples obtained from the same patients before and after immunotherapy (Supplementary Figure S5c). Whereas the majority of patient samples did not show any difference in the overall expression levels of I κ B ζ pre- or post-therapy, some tumors exhibited profound changes. Overall, we observed a trend where patients who did not respond to immunotherapy displayed upregulated I κ B ζ protein expression following immunotherapy, whereas patients with a complete response displayed decreased levels of I κ B ζ after immunotherapy. Importantly, none of the analyzed patients showed upregulated I κ B ζ expression in the presence of a complete immunotherapy response, or vice versa a downregulation of I κ B ζ expression together with a failed therapy response. Together, these findings suggest a link between I κ B ζ expression and immunotherapy responses.

Additionally, we expanded our analyses on immunotherapy responses using the murine melanoma cell line YUMM1.7, which endogenously expresses I κ B ζ . Treatment with α -PD-1 antibodies did not change the overall protein levels of I κ B ζ in the tumor, implying that I κ B ζ expression is not directly altered by immunotherapy. Rather, we propose that I κ B ζ expression modulates immunotherapy responses. However, the mechanisms upregulating I κ B ζ expression in melanoma, especially upon different therapies, remain unclear and should be addressed in future research.

Concerning a potential correlation between I κ B ζ target genes and immunotherapy responses, multiple reports have shown a correlation between single I κ B ζ target genes and ICB responses. For example, a previous report discovered that high levels of CXCL10 correlate with the response to immunotherapy¹. Furthermore, increased levels of tumor-derived *IL6* were detectable only in patients with ICB resistance but not in responding patients², whereas the blockade of IL-6 was shown to synergize with immunotherapy, at least in kidney, breast, and bladder cancer³.

Furthermore, we conducted a correlation analysis of I κ B ζ target gene expression and immunotherapy responses using data from a previous publication⁴ (Figure 1 for Reviewers' use). To this end, we used bulk RNA sequencing data from melanoma patients before immunotherapy⁴. We grouped patients with high levels of *CXCL10*, *CXCL9*, or *CCL5* (altered group) and compared their immunotherapy response to patients with normal expression levels of the depicted genes (Figure 1a for Reviewers' use). Of 121 patients, 9 patients exhibited elevated levels of *CXCL10* or *CXCL9* in the tumors, and this patient group tended to have a better immunotherapy response and prolonged overall survival (Figure 1b and c for Reviewers' use). However, as the data could only be retrieved from 9 patients, significance was not reached. Moreover, as the data derived from bulk sequencing, chemokine expression in tumor versus immune cells could not be distinguished. Consequently, we did not incorporate this data into the manuscript.

Figure 1 for Reviewers' use. Correlation of CXCL10 and CXCL9 expression in melanoma pre-treatment and correlation with immunotherapy responses. The data was retrieved and analyzed from cbiportal.org and originally published by Liu et al.⁴. *CXCL9*, *CXCL10*, and *CCL5* served as a combined query. **A.** Percentage of melanoma patients with high mRNA levels of *CXCL9*, *CXCL10*, and *CCL5*, graphically illustrated by oncoprint. Of note, *CCL5* expression was not upregulated. Therefore, it was excluded from further analysis. **B.** Immunotherapy responses of the altered patient's group (*CXCL10* and *CXCL9* high), compared to the unaltered patient's group. Shown are the immunotherapy responses based on RECIST1.1 criteria. **C.** Kaplan-Meier curve, showing the overall survival of the altered (*CXCL9* and *CXCL10* high) group (in red) and the unaltered group (in blue).

Authors should also test whether I κ B ζ expression is an independent or better predictor than other correlates of response, such as increased T cell infiltrates and immune signatures previously described. Co-expression of I κ B ζ and STAT3-P or p65 could also improve the predictive potential.

As suggested by the Reviewer, we performed staining of phosphorylated STAT3 (Y705, pSTAT3) and correlated its expression pattern to the presence or absence of tumor-derived I κ B ζ expression (Figure 8b). Interestingly, we detected a strong correlation between pSTAT3-positive tumor cells and I κ B ζ expression, especially in patients exhibiting immunotherapy resistance. This result validates our previous findings that I κ B ζ expression promotes the activity of STAT3, and agrees with earlier reports identifying hyperactivated STAT3 as a hallmark of immunosuppression and accelerated tumor growth⁵. We did not perform further analysis of p65 levels in melanoma patients, as total p65 levels do not reflect its activity, and detection of its subcellular localization might be heterogeneous within the tumor due to its oscillating activity.

A correlation of CD8⁺ T-cell infiltration, I κ B ζ expression, and ICB response was already assessed by us in the first version of the manuscript and is presented in Figures 5a and 5b. Our analysis revealed a very strong inverse correlation between CD8⁺ T-cell infiltrates and I κ B ζ protein expression. This finding implies that I κ B ζ expression could serve as an independent second predictor of immunotherapy responses alongside CD8⁺ T-cell infiltration.

Furthermore, we performed α -PD-1L staining of the human melanoma cohort, which is still considered a predictor of immunotherapy responses⁶. However, our data indicate no correlation between the immunotherapy response and the TPS (tumor proportion score), which quantifies linear membrane staining of PD-1L only on tumor cells. Therefore, we suggest that I κ B ζ expression might be a better predictor for ICB responses than quantification of PD-1L⁺-tumor cells. The new data is added to the manuscript as Supplementary Figure S5b.

2. The impact of I κ B ζ on ICB response is addressed by its overexpression in B16-F10 cells treated with anti-PD-1. However, to demonstrate its potential as therapeutic target, I κ B ζ downregulation in anti-PD-1 resistant cells (e.g. YUMM1.7) would be more relevant.

We thank the reviewer for this valuable point. As shown in Figure 1, YUMM1.7 cells constitutively express endogenous I κ B ζ . Therefore, we generated *Nfkbiz* KO YUMM1.7 cells and repeated the mouse model using immunocompetent C57BL/6 mice. Consistent with our findings in D4M-3A cells, knockout of I κ B ζ severely impaired the overall tumor growth rate of YUMM1.7 cells (Supplementary Figure S4b). Moreover, we detected diminished *I11a* and *I16* expression and elevated *Cxcl9*, *Cxcl10*, and *Ccl5* expression in *Nfkbiz* KO YUMM1.7 cells compared to control cells, thereby validating our previous results in I κ B ζ -overexpressing B16-F10 and *Nfkbiz* KO D4M-3A cells (Supplementary Figure S5d).

Subsequently, we evaluated immunotherapy responses in control and *Nfkbiz* KO YUMM1.7 tumors (Figure 7). As the knockout of I κ B ζ alone already impaired tumor growth, immunotherapy only mildly increased tumor growth inhibition (Figure 7c). However, in contrast to control YUMM1.7 cells, which were resistant to immunotherapy, we detected strong tumor growth inhibition of *Nfkbiz* KO YUMM1.7 tumors. Furthermore, α -PD-1 treatment induced CD8⁺ T-cell and NK-cell infiltration only in *Nfkbiz* KO tumors, but not in the controls (Figure 7d). This was accompanied by a strong upregulation of *Cxcl9*, *Cxcl10*, and *Ccl5* expression in the α -PD-1-treated *Nfkbiz* KO YUMM1.7 tumors (Figure 7e). Together this data validates melanoma-derived I κ B ζ as a key factor promoting immunotherapy resistance through the repression of chemokine-mediated recruitment of cytotoxic immune cells.

It would be interesting as well to address the changes in the TME driven by I κ B ζ blockade together with anti-PD-1.

As suggested by the Reviewer, we performed an extended analysis of the tumor microenvironment by flow cytometry of IgG- and α -PD-1-treated control and *Nfkbiz* KO YUMM1.7 tumors. In addition to the increased recruitment of CD8⁺ T-cells and NK cells, we detected increased infiltration of dendritic cells and monocytes into *Nfkbiz* KO tumors (Figure 7d). These changes were already detectable in IgG-treated tumors and further increased upon α -PD-1 antibody treatment. A recent publication uncovered pro-inflammatory monocytes that assist dendritic cells to prime and activate CD8⁺ T-cells in the tumor microenvironment⁷. Based on these findings, we propose that the increased infiltration of dendritic cells and pro-inflammatory monocytes into α -PD-1-treated *Nfkbiz* KO tumors could reflect the re-establishment of an anti-tumoral microenvironment upon I κ B ζ deletion. We acknowledge that a full characterization and possible modulation of the TME would provide deeper insights into how I κ B ζ -expressing tumors alter the tumor microenvironment. However, these investigations are beyond the scope of the current study and, due to the substantial additional work, would significantly extend the required revision time.

Minor

1. Authors do not justify why p50 is not included in the mechanistic assays when it is one of the known transcription factors that can interact with I κ B ζ together with p65 or STAT3.

We thank the reviewer for this point. We concentrated on p65, as p65 chromatin immunoprecipitation assays were already established in the lab. However, we conducted additional experiments with p50 that we have now incorporated into the manuscript. Figure 8c shows impaired chromatin localization of p50 in *NFKBIZ* knockdown LOX-IMVI cells. Furthermore, we performed a p50 chromatin immunoprecipitation assay using LOX-IMVI control and *NFKBIZ* knockdown cells. Consistent with our findings for p65 and STAT3, we detected impaired binding of p50 to I κ B ζ target gene promoters. The data is now presented in Supplementary Figure S7c. Thus, I κ B ζ deficiency affects the transcription factor function of p65 and p50, which aligns well with data reported in other cellular systems⁸.

2. The RNAseq analysis of I κ B ζ KO cells showed little overlap of differentially expressed genes between the 2 cell lines studied (LOX-IMVI and D4M-3A). Indeed, only one gene in common is related with immune pathways, which are the functions of other 18 genes in common? All of that should be discussed.

The overlap of common I κ B ζ target genes in melanoma and immune pathways is indeed bigger than initially anticipated, primarily due to the stringent filters we had initially set (fold change \geq 2, p-value \leq 0.05). However, when we loosened the filtering criteria (fold change \geq 1.5, p-value \leq 0.1), we found a broader overlap, especially in immune-related pathways and inflammation (Supplementary Figure S2b). In addition, certain genes were only expressed in LOX-IMVI or D4M-3A cells, which explains their absence in the initial overlap analysis. A complete list of conserved I κ B ζ target genes can be found in Table S1.

Moreover, we added a gene set enrichment analysis on the overlapping pathways derived from Figure 2 (Supplementary Figure S2a). This analysis revealed that common I κ B ζ target genes in melanoma are related to tumor growth and survival, immune evasion, inflammation, and metabolism. These findings support our conclusion that I κ B ζ regulates critical oncogenic pathways in melanoma. However, it should be noted that the functional annotation of these genes is very much dependent on the context and cell type. As we did not perform any further functional studies on these genes, we decided not to add this data to the

discussion to avoid potential overinterpretation. Further studies are needed to unravel the effect of I κ B ζ on other pathways including metabolism in melanoma.

Reviewer #3 (Remarks to the Author); expert in NF κ B signaling:

The manuscript from Kolb et al reports interesting results surrounding a role for I κ B ζ (hereafter I κ B ζ) in melanoma. As the authors note, I κ B ζ expression has not been studied extensively in solid tumors, so the results are important to better understand the oncogenic functions study this enigmatic member of the I κ B family. The authors utilize melanoma cell lines, corresponding cell line-derived tumor studies, and analysis of patient of samples to approach oncogenic mechanisms associated with I κ B ζ . Key reported data are: I κ B ζ protein levels are not correlated with mRNA levels, about 35% of melanoma tumors express I κ B ζ expression predicts poor outcomes, and I κ B ζ drives tumor growth and suppression of T cell recruitment and correspondingly resistance to immunotherapy. I κ B ζ modulates the recruitment of STAT3 and NF- κ B RelA to some promoters. For some tumors, there is a correlation between I κ B ζ expression and response to checkpoint inhibitors. While there are clearly some interesting results, some concepts are not well-developed and remain unclear.

We thank the reviewer for the thorough evaluation of our work and the valuable feedback. In response, we have now conducted new experiments providing additional insights into I κ B ζ function, especially on the mechanism of I κ B ζ -mediated gene repression in melanoma.

1) While cell line data do not show a correlation between I κ B ζ mRNA levels and protein, the authors should analyze TCGA melanoma datasets for I κ B ζ mRNA levels along with proposed target genes. Maybe human tumors show a correlation between I κ B ζ mRNA and protein levels and cell lines are not a good model for this analysis.

We performed RNAScope analyses to assess the relative abundance of *NFKBIZ* mRNA and its correlation to I κ B ζ protein using a tissue array containing 50 different melanoma tumors. The data, presented in Supplementary Figure S1h, did not reveal any correlation between *NFKBIZ* mRNA and I κ B ζ protein expression. Several tumors showed high copy numbers of *NFKBIZ* mRNA, yet no I κ B ζ protein was detectable. Furthermore, some tumors showed I κ B ζ protein expression, whereas only very little mRNA of *NFKBIZ* was detectable. This analysis validates our findings from cell culture experiments showing that mRNA and protein levels of I κ B ζ in melanoma do not correlate. Similarly, we could not reveal any significant correlation between *NFKBIZ* mRNA levels and I κ B ζ target gene expression, based on the TCGA data from Liu et al. ⁴. Therefore, we only provide this correlation analysis as Figure 2 for Reviewers' use.

Figure 2 for Reviewer's use. Correlation of *IL6*, *IL1A*, and *CXCL10* expression with *NFKBIZ* mRNA levels in melanoma (Liu et al. 2019). Correlation analysis was performed with cBioportal. As shown, Spearman correlation analysis retrieved no significant correlation.

2) Fig. 1 reports that MG132 stabilizes I κ B ζ protein levels, but for one of the cell lines MG132 increases the mRNA, and for another that is only a weak increase in protein levels. It was unclear the interpretation of the effect of the cap-dependent translation inhibitor.

Initially, we performed experiments using a 24-hour incubation time of MG132. Besides potential toxicities or off-target effects, other I κ B ζ -regulatory proteins might be influenced by prolonged proteasome inhibition and thereby indirectly affect mRNA levels of *NFKBIZ*. We have now conducted a refined analysis using a shorter, 4-hour treatment with MG132 and replaced the previous data from the 24-hour experiment (Figure 1c). Our new results confirmed that inhibition of the proteasome for 4 hours is sufficient to stabilize I κ B ζ protein levels, without altering *NFKBIZ* mRNA expression during this shorter incubation period.

In addition, to clarify the interpretation of the results with the cap-dependent translation inhibitor, we added data using actinomycin D, a small molecule inhibitor that blocks transcription (Supplementary Figure S1c). As expected, actinomycin D treatment quickly downregulated the mRNA levels of *NFKBIZ*, but not its protein levels, in two cell lines that constitutively express I κ B ζ . These findings further confirm that constitutive I κ B ζ expression in melanoma is predominantly regulated at the post-transcriptional or post-translational level rather than the transcriptional level.

3) In Fig. 1, how were the Kaplan-Meier results obtained? There was nothing in the main text about how the data were derived.

We added a more detailed description of the data and the generation of the Kaplan-Meier curves in the Methods section:

"Patient material, clinical data, and survival analysis. Tissue samples were obtained from the tissue biobanks of the University Medical Center Mainz and the West German Cancer Center Essen following their guidelines and approval of the local ethics committees (No. 837.226.05(4884) for Mainz and No. 11-4715 for Essen). Clinical characteristics of the patient cohort used to assess I κ B ζ levels in patients treated with ICB were stratified by clinical response criteria according to RECIST and are listed in Supplementary Table S4. Details on treatment and outcome specifics were recorded in an unidentifiable, pseudonymized format at the patient level. Biopsies for evaluation of immunotherapy responses and their correlation with I κ B ζ expression were obtained prior to the start of the immunotherapy. To detect I κ B ζ in melanoma samples from different stages and anatomical sites, we used a commercial tissue microarray from BioCat GmbH (Cat. ME1002b-BX). Samples were categorized into three groups: no detectable I κ B ζ protein expression (graded as 0), moderate I κ B ζ protein expression (graded as 1), or high overall expression

(graded as 2). I κ B ζ staining in infiltrating immune cells was neglected and excluded from the analysis. Subsequently, I κ B ζ levels were correlated with the clinical endpoints. The primary endpoints of this study were progression-free survival (PFS) and real-world tumor response following systemic therapy for metastatic stage IV upon adjuvant treatment failure. PFS was calculated from the start of initial treatment for metastatic stage IV until disease progression or death from any cause. Real-world tumor response was categorized as complete response (CR), partial response (PR), stable disease (SD), and progressive disease (PD) as described⁹.

We employed Kaplan–Meier survival plots to illustrate median progression-free survival probabilities. Survival curves were compared using a log-rank test. In all cases, two-tailed p-values were calculated and considered significant for $p < 0.05$. Survival analyses were conducted using the survminer R package (RStudio Version 1.3.1093).

4) Gene expression analysis using cell lines and knockdown or overexpression of I κ B ζ revealed previously identified targets such as IL-1b and IL-6. It was unclear relative to other targets as to their common regulation across the cell lines. Additionally, genes upregulated by I κ B ζ knockdown were not a focus of this section (and come up later). For example, Cxcl9, Cxcl10 – what were the effects on I κ B ζ knockdown/overexpression among the cell lines? And see comment above regarding TCGA analysis to correlate with the cell line data. Generally consistent with these results, I κ B ζ ChIPs at some of these promoters, including Cxcl10 (and see below).

Supplementary Figure S2d-g displays gene expression analysis of additional I κ B ζ target genes in various melanoma cell lines following overexpression or knockdown of *NFKB1Z*. To facilitate the evaluation of the conserved regulation of I κ B ζ target genes in the Supplementary Information, we modified the graphical illustration of Supplementary Figure S2 and grouped the data according to single genes (such as *Ccl5* or *Cxcl9*), rather than by cell line, similar as shown for *IL1B* and *IL6* in Figure 1. The data implies that I κ B ζ suppresses the expression of *CCL5*, *CXCL9*, and *CXCL10*, and induces *CXCL1* in several human and murine melanoma cell lines.

5) I κ B ζ was shown to promote cell growth/viability (although the relative contribution of either mechanism is not clear). I κ B ζ expression in B16 cells/tumors promotes tumor growth in C57/BL6 mice. In this regard, the authors should do the same study in immune-deficient mice to determine the impact of the endogenous immune system on tumor growth which was not addressed specifically, although implied.

We thank the reviewer for this valuable suggestion. To address this point, we repeated the steady-state analysis of the tumor growth of control and I κ B ζ -overexpressing B16-F10 cells in immunodeficient NOD/Scid mice from Janvier (NOD.CB17-Prkdcscid/scid/Rj) and added the data in Figure 4c and Figure 4f. Of note, this mouse strain lacks T-, B-, and NK cells, and is deficient in antigen-presenting cells. As expected, I κ B ζ -expressing tumors exhibited enhanced tumor cell growth, independent of the tumor microenvironment (Figure 4c). Furthermore, we detected increased *I11a* and *I16* mRNA levels, and downregulated *Ccl5*, *Cxcl9*, and *Cxcl10* expression in I κ B ζ -overexpressing tumors from NOD/Scid mice, confirming that the accelerated growth of I κ B ζ -overexpressing tumors is mainly driven by tumor-intrinsic effects (Figure 4f).

6) I κ B ζ expression suppressed the recruitment of CD8 T cells into the tumors, and I κ B ζ -expressing tumors showed reduced expression of Cxcl9, and 10. The question is whether the loss of Cxcl9 and 10 is intrinsic to the tumor and its level of I κ B ζ expression or due to lack of TME-derived

chemokines in the tumor microenvironment (see points above pt. 4 above, and points 8 and 9 below).

Our *in vitro* data on multiple melanoma cell lines showed that tumor-derived I κ B ζ suppresses the expression of various chemokines by directly binding to the respective gene loci (Figure 2h). Moreover, our steady-state analysis of control and I κ B ζ -expressing B16-F10 cells in immunodeficient NOD/Scid mice confirmed that these genes are simultaneously suppressed by I κ B ζ , even in the absence of T- or B-cells, or antigen-presenting cells (Figure 4c and 4f). Therefore, at least in steady-state, our data implies that I κ B ζ directly suppresses the expression of chemokines responsible for attracting T- and NK cells. Additionally, we performed immunofluorescence staining of CD45 (to discriminate tumor cells from immune cells) and CXCL9 in α -PD-1-treated control and I κ B ζ -overexpressing B16-F10 tumors at the endpoint (Figure 6h). We could validate a massive increase in infiltrating CD45⁺ immune cells and CXCL9 staining of α -PD-1-treated control B16-F10 tumors, which was absent in α -PD-1-treated I κ B ζ -overexpressing tumors. Interestingly, we detected CXCL9 expression in both, infiltrating immune cells and tumor cells in the α -PD-1-treated control group. These results suggest that I κ B ζ may repress chemokine expression in two different ways: either directly in the tumor cells or indirectly by suppressing the infiltration and possibly activation of chemokine-expressing immune cells. We added this aspect to the discussion.

7) Human patient data suggest an inverse correlation between I κ B ζ expression and response to immunotherapy for a portion of patients. Notably, there are both responders and non-responders among those patients with tumors that don't express I κ B ζ , so additional mechanisms must be at play to promote poor response to therapy.

We agree with the reviewer that other mechanisms might contribute to immunotherapy resistance apart from I κ B ζ . This could be because the activation of NK cells and cytotoxic T-cells might be compromised by alternative pathways as well. A lack of I κ B ζ protein expression does not necessarily imply that the patient is sensitive to immunotherapy. However, our data shows that tumor-derived I κ B ζ expression promotes immunotherapy resistance by suppressing the recruitment of NK cells and cytotoxic T-cells under steady-state conditions and upon immunotherapy.

Of note, single patients who display I κ B ζ expression in the tumor can still respond to immunotherapy. This observation might be explained by the fact that I κ B ζ lacks any enzymatic activity or DNA-binding ability by itself but rather serves as a bridging factor that recruits certain transcription factors and epigenetic modifiers to specific gene loci¹⁰. In agreement, we show that I κ B ζ promotes STAT3 and NF- κ B activity (Figure 8), and simultaneously recruits epigenetic modifiers such as HDAC3 and EZH2 to certain gene promoters, thereby mediating gene repression (Figure 9). As I κ B ζ strictly depends on the presence and activity of these effector molecules, mutations or aberrant expression of e.g. EZH2 and HDAC3 could lead to the abolishment of I κ B ζ 's oncogenic function. In agreement with this hypothesis, mutations of EZH2 have been described in multiple tumor entities^{11,12}. Thus, it will be interesting to investigate in future studies whether these rare patients, who are sensitive to immunotherapy but display constitutive I κ B ζ expression, exert mutations or functional impairments in HDAC3, EZH2, STAT3, or NF- κ B. We added this aspect to the discussion.

8) IκBζ expression in the B16 tumors suppressed the response to anti-PD-1. In these studies, anti-PD-1 promoted recruitment of T cells into the tumor and this was blocked with IκBζ tumor expression. Interestingly, it was the anti-PD-1 treatment that led to an increase of Cxcl9 and 10, and this was blocked by IκBζ expression. Regarding points above about repression of Cxcl9 and 10, are they suppressed in the cell lines or is it that in tumors, IκBζ suppresses T cell recruitment which leads to a lack of chemokines to induce their expression?

As discussed above, our experiments revealed that IκBζ expression suppresses tumor-derived chemokine expression in cell lines (Figure 2, Supplementary Figure S2) as well as in tumors derived from immunodeficient mice (Figure 4c and 4f). Immunofluorescence staining of α-PD-1-treated B16-F10 tumors revealed that at least CXCL9 expression is not exclusively derived from tumor cells but also from infiltrating immune cells. Therefore, melanoma-derived IκBζ seems to primarily repress T-cell-attracting chemokines directly in tumor cells and additionally enhances this effect by suppressing the expression of the same chemokines in the tumor microenvironment.

9) There is no explanation for the effects of IκBζ on STAT1/3 phosphorylation. Does IκBζ regulate JAK/STAT activity? Regarding the effect of IκBζ on the repression of Cxcl9 and 10 – does IκBζ promoter localization block the recruitment of STAT proteins and NF-κB to the promoter? Thus there is no explanation for how IκBζ suppresses Cxcl9 and 10 gene expression (and see points above).

We thank the reviewer for bringing up these very interesting questions concerning the mechanism of action of IκBζ, which are highly complex issues that warrant further investigation.

Concerning the effect of IκBζ on STAT1/3 phosphorylation, multiple mechanisms might explain this effect. The simplest explanation of the observed increase of pSTAT1 and pSTAT3 levels in IκBζ-overexpressing cells is the increased expression of tumor-intrinsic cytokines, such as IL-6, which directly induce Janus-dependent kinases (JAK) and STAT phosphorylation. Furthermore, it has been published that IκBζ can directly interact with STAT3¹³. Therefore, IκBζ might interfere with the interaction of dual-specificity phosphatases (DUSP), which are known to dephosphorylate STAT proteins, thereby inhibiting their dephosphorylation¹⁴. However, further molecular studies are needed to thoroughly dissect the crosstalk of IκBζ and STAT transcription factors, which we feel is beyond the scope of the present study.

Furthermore, we agree that IκBζ-mediated repression of *Ccl5*, *Cxcl9*, and *Cxcl10* deserves further investigations. Experiments using a STAT (Stattic) or NF-κB inhibitor (IKKβ inhibitor) could not abrogate the IκBζ-dependent suppression of these genes in IκBζ-expressing melanoma cell lines. We added this data as Supplementary Figures S8a and b.

Our data suggest that IκBζ-mediated gene repression could be mediated by alternative effector molecules. Several publications have shown that HDACs, especially HDAC3¹⁵⁻¹⁸, and EZH2^{19,20} can suppress *Ccl5*, *Cxcl9*, and *Cxcl10* expression in melanoma, contributing to immunotherapy resistance. In this context, we previously published that in keratinocytes, inhibition of EZH2 abrogates IκBζ-dependent gene expression²¹, and that IκBζ can directly interact with HDACs to mediate gene repression²². Thus, we propose that HDACs and/or EZH2 might mediate IκBζ-dependent gene repression in melanoma.

Indeed, we found that both, HDAC3 and EZH2, can interact with IκBζ in melanoma, leading to an IκBζ-dependent repression of gene expression. In detail, treatment with an HDAC3 or EZH2 inhibitor could abrogate IκBζ-mediated repression of *Ccl5* and *Cxcl10*, or *Cxcl10* and *Cxcl9*, respectively, in IκBζ-overexpressing B16-F10 cells (Figure 9a and 9b). Moreover, we detected a direct interaction between HDAC3 or EZH2 and IκBζ (Figure 9c). In agreement with these findings, overexpression of IκBζ in B16-F10 cells induced HDAC3 binding and EZH2-mediated trimethylation of H3K27 (H3K27me3) at the promoters of

those genes that were repressed by I κ B ζ (Figure 9d). Conversely, the knockdown of I κ B ζ in LOX-IMVI cells abrogated HDAC3 binding and H3K27 trimethylation at the promoter regions of *CCL5*, *CXCL9*, and *CXCL10* (Figure 9e). Thus, our findings indicate that I κ B ζ -mediated gene repression in melanoma cells relies on the direct recruitment of HDAC3 and EZH2 to distinct promoter sites, leading to their inaccessibility and silencing of gene expression.

Of course, we do not yet know which molecular mechanism dictates whether I κ B ζ promotes gene expression via STAT3 and NF- κ B or represses genes via HDAC3 and EZH2. We propose that post-translational modifications on I κ B ζ or specific regulatory elements in the distinct I κ B ζ target gene promoters might determine its overall function as a gene repressor or activator. Future investigations are needed to dissect the differences in I κ B ζ -mediated gene induction or repression in more detail. We added this point to the discussion.

10) Does I κ B ζ coIP with RelA and STAT3? I see the experiment described in the Methods, but don't see the description in the main text.

We did not perform Co-IP experiments of I κ B ζ and p65 or STAT3, as the interaction of I κ B ζ at least with STAT3 and p50 has been published before^{13,23}. Possibly, this was a misunderstanding in the Methods section, as we performed both, co-immunoprecipitation and chromatin immunoprecipitation assays. However, we only conducted Co-IPs of I κ B ζ using different I κ B ζ antibodies to validate our homemade antibody, whereas I κ B ζ , STAT3, p65, and p50 were assayed by chromatin immunoprecipitation analyses.

References

- 1 Reschke, R. *et al.* Immune cell and tumor cell-derived CXCL10 is indicative of immunotherapy response in metastatic melanoma. *J Immunother Cancer* **9**, doi:10.1136/jitc-2021-003521 (2021).
- 2 Hailemichael, Y. *et al.* Interleukin-6 blockade abrogates immunotherapy toxicity and promotes tumor immunity. *Cancer Cell* **40**, 509-523 e506, doi:10.1016/j.ccell.2022.04.004 (2022).
- 3 Huseni, M. A. *et al.* CD8(+) T cell-intrinsic IL-6 signaling promotes resistance to anti-PD-L1 immunotherapy. *Cell Rep Med* **4**, 100878, doi:10.1016/j.xcrm.2022.100878 (2023).
- 4 Liu, D. *et al.* Integrative molecular and clinical modeling of clinical outcomes to PD1 blockade in patients with metastatic melanoma. *Nat Med* **25**, 1916-1927, doi:10.1038/s41591-019-0654-5 (2019).
- 5 Zou, S. *et al.* Targeting STAT3 in Cancer Immunotherapy. *Mol Cancer* **19**, 145, doi:10.1186/s12943-020-01258-7 (2020).
- 6 Taube, J. M. *et al.* Association of PD-1, PD-1 ligands, and other features of the tumor immune microenvironment with response to anti-PD-1 therapy. *Clin Cancer Res* **20**, 5064-5074, doi:10.1158/1078-0432.CCR-13-3271 (2014).
- 7 Elewaut, A. *et al.* Cancer cells impair monocyte-mediated T cell stimulation to evade immunity. *Nature* **637**, 716-725, doi:10.1038/s41586-024-08257-4 (2025).
- 8 Daly, A. E., Yeh, G., Soltero, S. & Smale, S. T. Selective regulation of a defined subset of inflammatory and immunoregulatory genes by an NF-kappaB p50-IkappaBzeta pathway. *Genes Dev*, doi:10.1101/gad.351630.124 (2024).
- 9 Haist, M. *et al.* The Role of Treatment Sequencing with Immune-Checkpoint Inhibitors and BRAF/MEK Inhibitors for Response and Survival of Patients with BRAFV600-Mutant Metastatic Melanoma-A Retrospective, Real-World Cohort Study. *Cancers (Basel)* **14**, doi:10.3390/cancers14092082 (2022).
- 10 Feng, Y. *et al.* The central inflammatory regulator IkappaBzeta: induction, regulation and physiological functions. *Front Immunol* **14**, 1188253, doi:10.3389/fimmu.2023.1188253 (2023).
- 11 Qiu, J., Sharma, S., Rollins, R. A. & Paul, T. A. The complex role of EZH2 in the tumor microenvironment: opportunities and challenges for immunotherapy combinations. *Future Med Chem* **12**, 1415-1430, doi:10.4155/fmc-2020-0072 (2020).
- 12 Gallagher, S. J., Tiffen, J. C. & Hersey, P. Histone Modifications, Modifiers and Readers in Melanoma Resistance to Targeted and Immune Therapy. *Cancers (Basel)* **7**, 1959-1982, doi:10.3390/cancers7040870 (2015).
- 13 Wu, Z. *et al.* Nuclear protein IkappaB-zeta inhibits the activity of STAT3. *Biochem Biophys Res Commun* **387**, 348-352, doi:10.1016/j.bbrc.2009.07.023 (2009).
- 14 Singh, M. K., Altameemi, S., Lares, M., Newton, M. A. & Setaluri, V. Role of dual specificity phosphatases (DUSPs) in melanoma cellular plasticity and drug resistance. *Sci Rep* **12**, 14395, doi:10.1038/s41598-022-18578-x (2022).
- 15 Woods, D. M. *et al.* HDAC Inhibition Upregulates PD-1 Ligands in Melanoma and Augments Immunotherapy with PD-1 Blockade. *Cancer Immunol Res* **3**, 1375-1385, doi:10.1158/2326-6066.CIR-15-0077-T (2015).
- 16 Liu, D. *et al.* The Circadian Clock Component RORA Increases Immunosurveillance in Melanoma by Inhibiting PD-L1 Expression. *Cancer Res* **84**, 2265-2281, doi:10.1158/0008-5472.CAN-23-3942 (2024).

- 17 Booth, L., Roberts, J. L., Poklepovic, A., Kirkwood, J. & Dent, P. HDAC inhibitors enhance the immunotherapy response of melanoma cells. *Oncotarget* **8**, 83155-83170, doi:10.18632/oncotarget.17950 (2017).
- 18 Li, L. *et al.* HDAC3 Inhibition Promotes Antitumor Immunity by Enhancing CXCL10-Mediated Chemotaxis and Recruiting of Immune Cells. *Cancer Immunol Res* **11**, 657-673, doi:10.1158/2326-6066.CIR-22-0317 (2023).
- 19 Li, C. *et al.* Mi-2beta promotes immune evasion in melanoma by activating EZH2 methylation. *Nat Commun* **15**, 2163, doi:10.1038/s41467-024-46422-5 (2024).
- 20 Wozniak, M. & Czyz, M. lncRNAs-EZH2 interaction as promising therapeutic target in cutaneous melanoma. *Front Mol Biosci* **10**, 1170026, doi:10.3389/fmolb.2023.1170026 (2023).
- 21 Müller, A. *et al.* The CDK4/6-EZH2 pathway is a potential therapeutic target for psoriasis. *J Clin Invest* **130**, 5765-5781, doi:10.1172/JCI134217 (2020).
- 22 Grondona, P. *et al.* Threonine Phosphorylation of I kappaBzeta Mediates Inhibition of Selective Proinflammatory Target Genes. *J Invest Dermatol* **140**, 1805-1814 e1806, doi:10.1016/j.jid.2019.12.036 (2020).
- 23 Trinh, D. V., Zhu, N., Farhang, G., Kim, B. J. & Huxford, T. The nuclear I kappaB protein I kappaB zeta specifically binds NF-kappaB p50 homodimers and forms a ternary complex on kappaB DNA. *J Mol Biol* **379**, 122-135, doi:10.1016/j.jmb.2008.03.060 (2008).